# Empower Words: DualGround for Structured Phrase and Sentence-Level Temporal Grounding

**Minseok Kang**
Yonsei University
louis0503@yonsei.ac.kr

**Minhyeok Lee**
Yonsei University
hydragon516@yonsei.ac.kr

**Minjung Kim**
LG Electronics
minjung12.kim@lge.com

**Donghyeong Kim**
Yonsei University
2donghyung87@yonsei.ac.kr

**Sangyoun Lee**
Yonsei University
syleee@yonsei.ac.kr

## Abstract

Video Temporal Grounding (VTG) aims to localize temporal segments in long, untrimmed videos that align with a given natural language query. This task typically comprises two subtasks: *Moment Retrieval* (MR) and *Highlight Detection* (HD). While recent advances have been progressed by powerful pretrained vision-language models such as CLIP and InternVideo2, existing approaches commonly treat all text tokens uniformly during cross-modal attention, disregarding their distinct semantic roles. To validate the limitations of this approach, we conduct controlled experiments demonstrating that VTG models overly rely on `[EOS]`-driven global semantics while failing to effectively utilize word-level signals, which limits their ability to achieve fine-grained temporal alignment. Motivated by this limitation, we propose DualGround, a dual-branch architecture that explicitly separates global and local semantics by routing the `[EOS]` token through a sentence-level path and clustering word tokens into phrase-level units for localized grounding. Our method introduces (1) token-role-aware cross modal interaction strategies that align video features with sentence-level and phrase-level semantics in a structurally disentangled manner, and (2) a joint modeling framework that not only improves global sentence-level alignment but also enhances fine-grained temporal grounding by leveraging structured phrase-aware context. This design allows the model to capture both coarse and localized semantics, enabling more expressive and context-aware video grounding. DualGround achieves state-of-the-art performance on both Moment Retrieval and Highlight Detection tasks across QVHighlights and Charades-STA benchmarks, demonstrating the effectiveness of disentangled semantic modeling in video-language alignment.

## 1 Introduction

Video Temporal Grounding (VTG) aims to localize segments in a video that correspond to a natural language query. VTG comprises two sub-tasks: *Moment Retrieval* (MR), which predicts the start and end timestamps of relevant moments, and *Highlight Detection* (HD), which assigns saliency scores to short video clips based on query relevance. Given their structural similarity and shared objective of grounding query-relevant content, recent approaches have explored joint training of MR and HD, particularly enabled by the QVHighlights dataset [10], which provides aligned annotations for both tasks. Furthermore, the use of pretrained vision-language models (VLMs), such as CLIP [22] and InternVideo2 [29], has improved query-video alignment through rich cross-modal representations.

39th Conference on Neural Information Processing Systems (NeurIPS 2025).

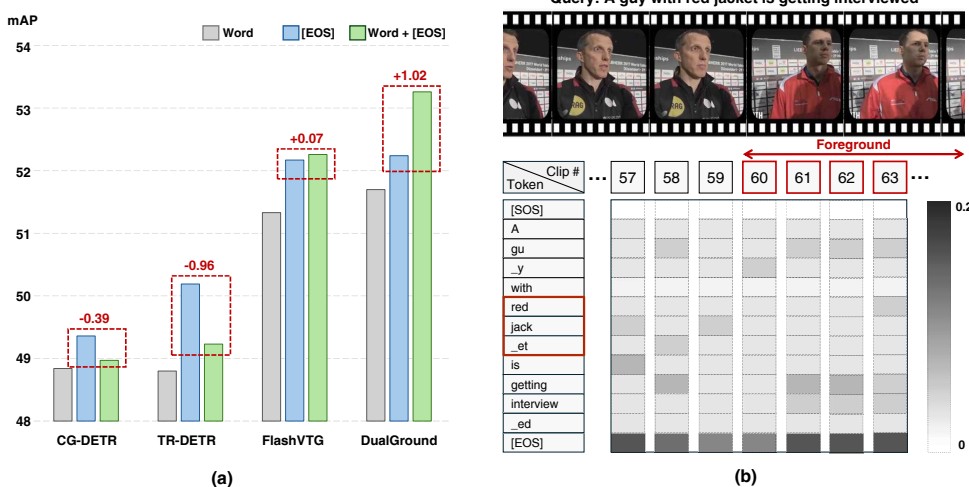

Figure 1: (a) Comparison of token input configurations (Word only, [EOS] only, Full (Word + [EOS])) on QVHighlights val set. DualGround reduces over-reliance on [EOS] and better utilizes word-level cues. (b) Cross-attention map from FlashVTG [1] showing weak attention to word tokens and dominant focus on [EOS], suppressing localized semantics.

Both CLIP and InternVideo2 tokenize input text queries with special tokens such as [SOS] and [EOS], which are positioned at the beginning and end of the token sequence, respectively. Crucially, the [EOS] token is designed not merely as a positional marker but as a summary representation of the entire sentence. It is trained to embed holistic semantic information derived from all preceding tokens, and thus becomes highly influential in downstream tasks.

Despite the inherent semantic differences between word-level tokens and the sentence-level [EOS] token, prior VTG models [24, 19, 18, 1] treat all text tokens uniformly during cross-modal attention. Since VTG aims to align video segments with the overall sentence intent, attention tends to concentrate on the [EOS] token, which encapsulates global semantics. This design, however, risks underutilizing localized word-level cues that are essential for fine-grained grounding.

We empirically investigate how current VTG models utilize textual representations through a comparison of three input configurations—(1) word tokens only, (2) the [EOS] token only, and (3) full sequences—on the QVHighlights validation set using InternVideo2 features. As shown in Figure 1, models achieve comparable or even superior performance when using only the [EOS] token compared to the full input. Attention visualizations further reveal that even for video clips unrelated to the input query, the model exhibits a predominant focus on the sentence-level [EOS] representation, while salient word tokens (e.g., "red jacket") that provide visually meaningful cues are largely underutilized. This phenomenon arises because the [EOS] token, produced by an off-the-shelf pretrained text encoder, is designed to summarize the entire sentence independently of the visual context. Consequently, the [EOS] token may fail to reflect textual cues that are visually salient and critical for accurate grounding. These observations underscore the importance of incorporating fine-grained word-level semantics for precise and context-aware moment localization. To substantiate this observation at scale, we provide a quantitative correlation analysis in **Appendix B.1**, demonstrating that prior VTG models exhibit consistently high alignment between the [EOS] and word-token attentions.

These findings reveal that existing VTG models are biased toward global sentence-level semantics, indicating the need for an approach that can more accurately ground videos where such bias hinders fine-grained alignment. To address this issue, we introduce a dual-branch architecture that jointly models global and local textual semantics for robust video-text alignment. This design retains the strong grounding capabilities of the sentence-level representation while incorporating a phrase-level path that clusters contextually coherent words into semantically meaningful units. The dual-path

structure allows the model to balance coarse global alignment and fine-grained local interactions, capturing nuanced word dependencies that are often diluted in flat token sequences.

Within the **sentence-level path**, we adopt Adaptive Cross Attention (ACA) to strengthen alignment between the sentence embedding and video clips. ACA incorporates learnable dummy tokens to absorb irrelevant attention, guiding semantically aligned clips to focus on the `[EOS]` token. This strategy mitigates the limitations of single-token attention and promotes stable sentence-level grounding with reduced interference from noisy textual inputs.

For the **phrase-level path**, we cluster word tokens into semantically coherent phrases based on their representational correlation in the feature space. Given that word semantics are context-dependent and emerge through interactions with neighboring words, phrase-level abstraction offers a more coherent representation for aligning with visual content. Inspired by recent multimodal reasoning researches [7, 20], we use these phrase approach as structured intermediate units that support fine-grained alignment. Initial phrase groupings are formed using a Recurrent Phrase Generator (RPG), which composes each phrase by attending over word tokens, conditioned on global semantics and prior phrase context. These groupings are then refined through a Slot Attention module, which disentangles overlapping meanings and enhances semantic purity through iterative updates.

Unlike prior works that treat textual features as flat sequences, our model explicitly captures interactions between each phrase and each video clip. We compute a dense phrase-clip context embedding via Hadamard product between projected phrase and video embeddings, followed by temporal self-attention to maintain consistency across time. This design enables the model to reason about semantic relevance at a fine temporal resolution. By unifying global and localized semantics in a structured manner, our model achieves more precise and context-aware alignment between language and video. This joint modeling not only improves retrieval accuracy but also enhances robustness across varying query complexities.

Building upon these observations and designs, our main contributions can be summarized as follows:

- We empirically identify a strong bias in existing VTG models toward the global `[EOS]` representation, which leads to underutilization of word-level semantics crucial for fine-grained grounding.

- We propose **DualGround**, a dual-path architecture that jointly models global sentence-level and localized phrase-level semantics, enabling balanced and context-aware video-text alignment.

- Our approach achieves precise and robust grounding by integrating dual-level textual semantics, achieving state-of-the-art performance on the QVHighlights and Charades-STA benchmarks.

## 2 Related Work

**Video Temporal Grounding.** VTG has been extensively studied through its two core sub-tasks: moment retrieval (MR) and highlight detection (HD). Early MR approaches can be categorized into either proposal-based or proposal-free paradigms. Proposal-based methods [6, 33, 37] first generate candidate temporal segments—typically via sliding windows or anchor mechanisms—and then rank them according to their relevance to the query. While effective, these methods often suffer from redundancy and coarse temporal boundaries. In contrast, proposal-free methods [10, 20] directly regress start and end timestamps or use attention-based localization, allowing end-to-end optimization and more flexible fine-grained reasoning. As a result, proposal-free frameworks have become dominant due to their efficiency and compositional flexibility.

A major milestone for unified VTG research was the introduction of the QVHighlights dataset [10], which provides aligned annotations for both MR and HD tasks over shared video–query pairs. This dataset enabled models to jointly optimize coarse-grained temporal localization and fine-grained clip-level saliency estimation, bridging the two previously disjoint tasks and facilitating cross-task supervision.

Building on this foundation, recent VTG models increasingly adopt DETR frameworks [10, 18, 19, 24, 26, 9, 8], where learnable decoder queries replace heuristic proposals to achieve end-to-end training and global reasoning. However, such methods rely on a limited number of decoder queries,

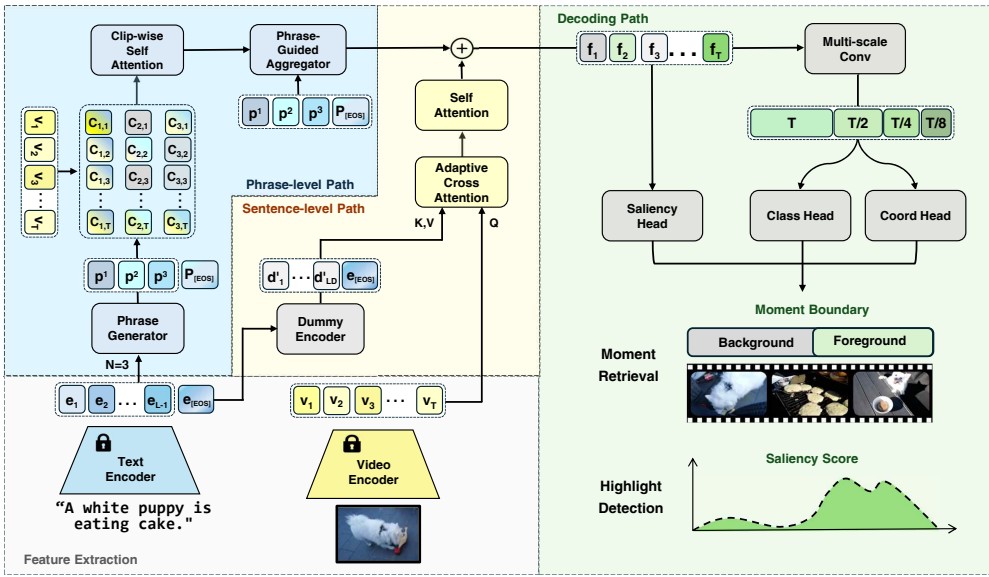

Figure 2: Overall architecture of DualGround. The model processes sentence-level semantics and phrase-level signals through separate pathways to capture both global intent and localized context. These two representations are then fused and used to perform moment retrieval and highlight detection with fine temporal precision.

restricting temporal granularity and making it difficult to capture short or densely overlapping events. To address these limitations, follow-up studies have introduced multi-scale temporal modeling (e.g., R2-Tuning [13] and FlashVTG [1]), which construct temporal pyramids or hierarchical representations to improve localization accuracy across diverse segment durations. This design paradigm draws inspiration from multi-scale feature encoding methods in temporal action localization, as in Action-Former [35]. Other lines of research enrich the multimodal representation space by incorporating additional modalities, including audio signals [14, 4, 3, 25] or by leveraging external knowledge from large language models (LLMs) [27, 16, 17] to improve grounding generalization and cross-domain robustness.

**Vision–Language Foundation Models.** Following the release of QVHighlights, VTG has entered a new stage driven by large-scale vision–language pretraining. The success of CLIP [22] demonstrated that contrastive multimodal representation learning could effectively align visual and textual semantics in open domains. Subsequent video–language models such as Video-LLaMA [36] and InternVideo2 [29], further extended this paradigm to spatiotemporal contexts, enabling generic video encoders to serve as universal feature extractors for VTG. By leveraging these pretrained representations, recent works have achieved remarkable transferability and data efficiency on downstream MR and HD benchmarks. This transition toward vision–language foundation model-based VTG has shifted the focus from purely architectural innovations to representation learning and cross-modal alignment quality, paving the way for unified and scalable grounding frameworks.

**Text-Centric Approach.** Complementary to architectural advances, recent studies have explored improving textual representations for accurate grounding. Woo et al. [30] proposed a holistic query understanding framework that employs a global text anchor to regulate visual attention, demonstrating the importance of sentence-level semantics in filtering irrelevant clips. Keyword-DETR [26] instead emphasizes visually salient keywords through token-level attention, highlighting the role of linguistically informative words in temporal grounding. These works share our motivation to strengthen textual cue utilization for precise video–text alignment.

# 3 Method

## 3.1 Model Overview

An overview of **DualGround** is provided in **Figure 2**. DualGround adopts a dual-path framework that integrates sentence-level and phrase-level semantics for video temporal grounding. **Section 3.2** introduces the sentence-level path, which leverages the [EOS] token to capture global alignment with the video. **Section 3.3** presents the phrase-level path, which clusters word tokens into localized phrases and models their interaction with the video. The decoding module that fuses these signals for moment retrieval and highlight detection is described in **Section 3.4**. Finally, the overall training objectives are detailed in **Section 3.5**.

## 3.2 Sentence-Level Path

In our design, we isolate the sentence-level representation using only the [EOS] token, resulting in a single-token key sequence. While this token captures global semantics, such a minimal representation is inherently incompatible with standard attention mechanisms, which rely on multiple key tokens to compute contrastive similarity distributions. To address this, we incorporate the Adaptive Cross Attention (ACA) mechanism introduced in CG-DETR [18], appending learnable dummy tokens to the key sequence. These dummy tokens are conditioned on the sentence and act as semantic attractors for video clips that are weakly aligned with the query, absorbing their attention and preventing interference with the alignment signal encoded in the [EOS] token. Conversely, clips that are semantically relevant to the query attend more directly to the [EOS] token, enabling sharp and robust global alignment. This design allows our model to simulate full-sequence sentence-level attention using only a compact representation, reducing reliance on noisy or less informative word-level tokens.

**Dummy-Enhanced Sentence Attention.** We represent the sentence-level embedding as the [EOS] token $e_{\texttt{[EOS]}} \in \mathbb{R}^d$, and introduce $L_d$ learnable dummy tokens $\{d_1, d_2, \ldots, d_{L_d}\} \in \mathbb{R}^d$. We stack them to form $D = \{d_i\}_{i=1}^{L_d} \in \mathbb{R}^{L_d \times d}$. To contextualize them, we concatenate $D$ and the [EOS] token to obtain $E = [D;\ e_{\texttt{[EOS]}}] \in \mathbb{R}^{(L_d+1) \times d}$, and pass this sequence through a lightweight Transformer encoder $f_{\text{enc}}$, which corresponds to the dummy encoder illustrated in Fig. 2. From the encoder output, we retain the first $L_d$ rows as the updated dummy embeddings $D' = \{d'\}_{i=1}^{L_d}$, and append the original $e_{\texttt{[EOS]}}$ to construct the attention input sequence $E' = [D';\ e_{\texttt{[EOS]}}] \in \mathbb{R}^{(L_d+1) \times d}$, where the [EOS] token is placed at the final index.

To compute cross-attention, video features $V = \{v_i\}_{i=1}^T \in \mathbb{R}^{T \times d}$ are projected into queries $Q = \{q_i\}$, and $E'$ is projected into keys $K = \{k_j\}$ and values $U = \{u_j\}$ using learnable linear layers. The attention weight for the $i$-th video clip is computed with respect to the [EOS] token's key vector as:

$$\alpha_i = \text{softmax}\left(\frac{q_i \cdot k_j}{\sqrt{d}}\right)\Bigg|_{j=L_d+1}, \quad \text{ACA}(v_i) = \alpha_i \cdot u_{L_d+1} \tag{1}$$

Here, $q_i$ is the query vector for the $i$-th video clip, and $k_j$, $u_j$ are the key and value vectors at position $j$. This setup allows each video clip feature $v_i$ to selectively attend to sentence-level semantics encoded in the [EOS] token, while dummy tokens act as attention sinks for noisy or irrelevant content. To enhance temporal coherence, we apply a stack of self-attention layers along the clip (temporal) dimension, producing the final sentence-guided video representation $V^s \in \mathbb{R}^{T \times d}$.

## 3.3 Phrase-Level Path

**Recurrent Phrase Generation.** To generate initial phrase representations, we aim to cluster the input sentence into a fixed number $N$ of semantically coherent units. We design a Recurrent Phrase Generation (RPG) module that incrementally composes phrases by attending over word tokens, conditioned on both the global sentence semantics and the previously generated phrases. Inspired by the sequential phrase grouping strategy in LGI [20], this formulation helps form contextually coherent and robust phrase-level representations. In addition, to encourage the grouping of adjacent words, we inject positional embeddings into the word tokens during phrase composition.

Let $\{e_1, \ldots, e_{L-1}\} \in \mathbb{R}^{(L-1) \times d}$ denote the word-level embeddings excluding the [EOS] token. We generate $N$ initial phrase representations in a recurrent manner, where each phrase is computed by using a guide vector to softly aggregate word-level embeddings. At each step $n$, the guide vector $g^{(n)}$ is constructed from the sentence-level embedding $e_{[EOS]}$ and the previously generated phrase $p_{i}^{(n-1)}$. For the first phrase —where no previous phrase exists— we instead use a zero-initialized placeholder vector. The transformation $\phi(\cdot)$ used to produce the guide vector is implemented as a lightweight MLP followed by a GELU activation. Once the guide vector is computed, it attends to the word tokens via scaled dot-product attention to produce the corresponding phrase embedding. The resulting phrase set is denoted as $P_i = \{p_{i}^{(1)}, \ldots, p_{i}^{(N)}\} \in \mathbb{R}^{N \times d}$:

$$p_{i}^{(n)} = \sum_{l=1}^{L-1} \text{softmax}\left(\frac{g^{(n)} \cdot e_l}{\sqrt{d}}\right) \cdot e_l, \quad \text{where } g^{(n)} = \begin{cases} \phi\left(W_q^{(1)} e_{[EOS]}, \mathbf{0}\right) & \text{if } n = 1 \\ \phi\left(W_q^{(n)} e_{[EOS]}, p_{i}^{(n-1)}\right) & \text{if } n \geq 2 \end{cases} \quad (2)$$

**Phrase Refine & Global Token Reconstruction.** These initial groupings are refined via a Slot Attention based module. While the initial clustering captures coarse semantic groupings, it may not fully disentangle overlapping or noisy meanings due to its limited capacity and sequential generation. To address this, we adopt a refinement mechanism that allows phrase embeddings to be iteratively updated in a more context-aware manner.

Slot attention [15] is particularly well-suited for refining initial phrase representations, as it treats each phrase embedding as a latent slot that selectively aggregates semantically aligned word-level features. However, its effectiveness can be sensitive to how the slots are initialized, making informed initialization important for stable refinement. Since our framework already generates context-aware phrase embeddings through sequential clustering, it naturally provides reliable initialization for slot attention, mitigating the sensitivity to slot quality. We then apply a slot-attention layer, where each phrase attends to word tokens treated as key-value inputs. The module employs slot-wise softmax followed by input-wise normalization to enable semantically coherent refinement without the need to discover clusters from scratch.

Let the refined phrase set be denoted as $P = \{p^{(1)}, \ldots, p^{(N)}\} \in \mathbb{R}^{N \times d}$. To further promote global coherence and inter-slot interaction, we append a learnable token $P_{[EOS]}$ to the phrase set and pass $[P; P_{[EOS]}]$ through a lightweight self-attention transformer block. Although the phrase-level path primarily models localized semantics, we introduce $P_{[EOS]}$ to explicitly consolidate the global meaning from phrase-level cues, enabling the model to maintain sentence-level semantics without relying on the $e_{[EOS]}$. As detailed in **Phrase-Guided Aggregation**, $P_{[EOS]}$ summarizes the overall phrase semantics and produces importance weights for aggregating the phrase-clip context.

**Phrase-Clip Context.** We model the semantic relevance between each phrase and video clip through a phrase-conditioned interaction implemented as a Hadamard product over projected representations. Given the refined phrase set $P \in \mathbb{R}^{N \times d}$ and video features $V \in \mathbb{R}^{T \times d}$, this process yields the phrase-clip context embeddings.

$$C = f_{\text{ctx}}(f_p(P) \odot f_v(V)) \in \mathbb{R}^{N \times T \times d} \quad (3)$$

All functions $f_p$, $f_v$, and $f_{\text{ctx}}$ are implemented as MLPs with GELU activations, enabling expressive modeling of interactions. To ensure temporal consistency, we apply stacked self-attention over clip dimension $T$ within each phrase stream.

**Phrase-Guided Aggregation.** The final step of our phrase-level path aggregates the phrase-clip context embedding $C \in \mathbb{R}^{N \times T \times d}$ into a unified phrase-guided representation $V_p \in \mathbb{R}^{T \times d}$, where each clip feature $v_{p,t}$ integrates information across all phrases.

We compute the semantic importance of each phrase by measuring its similarity to the reconstructed sentence-level token $P_{[EOS]} \in \mathbb{R}^d$. These attention weights are then used to aggregate across phrases for each time step $t$:

$$v_{p,t} = \sum_{n=1}^{N} \text{softmax}\left(\frac{\langle W_q P_{[EOS]}, W_k p^{(n)} \rangle}{\sqrt{d}}\right) \cdot C_{n,t} \quad (4)$$

This aggregation process allows the model to emphasize different phrases depending on their alignment with the global sentence-level intent, yielding a compositional representation that reflects phrase-aware relevance at each clip.

## 3.4 Decoding Path for Temporal Grounding

We follow the decoding strategy proposed in FlashVTG [1] and R2-Tuning [13], which replaces standard DETR-style decoding [10] with a multi-scale prediction framework. Unlike DETR, which predicts moment spans using a fixed number of learnable queries, our method directly performs predictions over the fused video-text feature $F = V_s + V_p = \{f_i\}_{i=1}^T \in \mathbb{R}^{T \times d}$, which is processed through a temporal feature pyramid constructed via stacked 1D convolutions. This design enables the model to capture moments of varying durations more effectively by making predictions at multiple temporal resolutions.

Moment boundaries are predicted at each scale using a shared prediction head. The multi-scale outputs are then concatenated and passed through classification and regression heads to produce moment confidence scores and normalized start/end points. Highlight detection is performed at the base resolution using a saliency scoring head that combines global and local context through Hadamard interaction.

## 3.5 Training Objectives

We adopt standard training objectives widely used in VTG literature. The overall loss consists of three components for Moment Retrieval (MR), Highlight Detection (HD), and Phrase-Level supervision.

**Moment Retrieval Loss.** We employ a classification loss (Focal loss [12]) and a boundary regression loss (L1) to supervise the moment prediction. The total moment retrieval loss is defined as $\mathcal{L}_{\text{mr}} = \mathcal{L}_{\text{cls}} + \mathcal{L}_{\text{reg}}$.

**Highlight Detection Loss.** Following prior work, the highlight detection loss is defined as the sum of four components: the ranking loss and contrastive loss over the clip-level saliency scores $S$, and the ranking loss and contrastive loss over the sentence-level attention weights $\alpha$. The overall highlight detection loss is expressed as $\mathcal{L}_{\text{hd}} = \mathcal{L}_{\text{rank}}^{(S)} + \mathcal{L}_{\text{contrast}}^{(S)} + \lambda_{\text{attn}}(\mathcal{L}_{\text{rank}}^{(\alpha)} + \mathcal{L}_{\text{contrast}}^{(\alpha)})$.

**Phrase-Level Loss.** To ensure that the phrase representations are both semantically disentangled and consistent with the overall sentence representation, we introduce a phrase-level objective $\mathcal{L}_{\text{phrase}}$ composed of two complementary terms.

**(1) Distinct Query Attention (DQA) Loss.** We encourage semantic diversity across phrases by regularizing their attention distributions to remain orthogonal. Let $A \in \mathbb{R}^{B \times N \times (L-1)}$ denote the attention weights over $(L-1)$ word tokens for $N$ phrases across a batch of size $B$. The DQA loss is defined using the Frobenius norm $\| \cdot \|_F$ as:

$$\mathcal{L}_{\text{DQA}} = \frac{1}{B} \sum_{i=1}^{B} \left\| A_i A_i^\top - r \cdot I \right\|_F^2 , \tag{5}$$

where $r$ is a scaling coefficient that controls the strength of self-correlation along the diagonal, determining how strictly each phrase attention is encouraged to be distinct.

**(2) EOS Reconstruction Loss.** To maintain alignment between phrase-derived representations and the sentence-level semantics, we introduce a reconstruction objective that aligns the reconstructed global token $P_{\texttt{[EOS]}}$ with the original $e_{\texttt{[EOS]}}$ embedding. Here, $\tau$ is a temperature hyperparameter and $\cos(\cdot, \cdot)$ denotes cosine similarity. We employ an InfoNCE [21] loss:

$$\mathcal{L}_{\text{EOS}} = - \log \frac{\exp(\cos(P_{\texttt{[EOS]}}, e_{\texttt{[EOS]}}^+)/\tau)}{\sum_{j=1}^{B} \exp(\cos(P_{\texttt{[EOS]}}, e_{\texttt{[EOS]}}^j)/\tau)} \tag{6}$$

**Total Loss.** The final training loss is computed as a weighted sum of three components: the moment retrieval loss $\mathcal{L}_{\text{mr}}$, the highlight detection loss $\mathcal{L}_{\text{hd}}$, and the phrase-level supervision loss, which includes the distinct query attention loss $\mathcal{L}_{\text{DQA}}$ and the $\texttt{[EOS]}$ reconstruction loss $\mathcal{L}_{\text{EOS}}$. Formally, the total loss is given by $\mathcal{L}_{\textbf{total}} = \lambda_{\textbf{mr}}\mathcal{L}_{\textbf{mr}} + \lambda_{\textbf{hd}}\mathcal{L}_{\textbf{hd}} + \lambda_{\textbf{phrase}}(\mathcal{L}_{\textbf{DQA}} + \mathcal{L}_{\textbf{EOS}})$, where each $\lambda$ controls the relative weight of the corresponding term.

Table 1: Video moment retrieval (MR) and highlight detection (HD) results on QVHighlights **Test** split. **Bold**: best overall, Underline: second best overall. SF+C denotes CLIP text features with SlowFast and CLIP video features; IV2 denotes InternVideo2 features for both modalities.

| Method | Backbone | R1@0.5 | R1@0.7 | mAP | mAP@0.5 | mAP@0.75 | VG-mAP | VG-Hit@1 |
|---|---|---|---|---|---|---|---|---|
| CG-DETR [18] | SF+C | 65.43 | 48.38 | 42.86 | 64.51 | 42.77 | 40.33 | 66.21 |
| TR-DETR [24] | SF+C | 64.66 | 48.96 | 42.62 | 63.98 | 43.73 | 39.91 | 63.42 |
| UVCOM [31] | SF+C | 63.55 | 47.47 | 43.18 | 63.37 | 42.67 | 39.74 | 64.20 |
| R2-Tuning [13] | C | 68.03 | 49.35 | 46.17 | 69.04 | 47.56 | 40.75 | 64.20 |
| FlashVTG [1] | SF+C | 66.08 | 50.00 | 48.70 | 67.99 | 47.59 | 41.07 | 66.10 |
| DualGround | SF+C | **68.20** | **51.72** | **49.02** | **69.23** | **47.71** | **41.15** | **66.30** |
| FlashVTG [1] | IV2 | 70.69 | 53.96 | 52.00 | 72.33 | 53.85 | **44.09** | **71.00** |
| DualGround | IV2 | **71.87** | **56.94** | **52.73** | **72.41** | **54.38** | 44.02 | 70.80 |

Table 2: Performance on QVHighlights **Validation** split using InternVideo2 features for fair comparison.

| Method | Backbone | R1@0.5 | R1@0.7 | mAP | mAP@0.5 | mAP@0.75 | VG-mAP | VG-Hit@1 |
|---|---|---|---|---|---|---|---|---|
| CG-DETR [18] | IV2 | 70.06 | 55.87 | 48.93 | 69.85 | 49.56 | 42.30 | 68.71 |
| TR-DETR [24] | IV2 | 71.72 | 55.93 | 48.93 | 70.87 | 50.14 | 43.74 | 70.84 |
| FlashVTG [1] | IV2 | 71.48 | 56.06 | 52.61 | 72.37 | 55.03 | 44.08 | 71.48 |
| DualGround | IV2 | **73.48** | **58.97** | **53.26** | **72.99** | **56.35** | **44.12** | **71.62** |

# 4 Experimental Results

## 4.1 Datasets & Evaluation Metrics

We evaluate our method on three benchmarks: **QVHighlights** [10], **Charades-STA** [6], and **TV-Sum** [23]. These cover both moment retrieval and highlight detection, across diverse domains including open-domain YouTube videos, indoor activities, and web videos.We adopt standard data splits and evaluation metrics used in prior works [19, 10, 13], including Recall@1 and mAP for moment retrieval, and mAP and HIT@1 for highlight detection. Detailed dataset statistics and metrics are provided in the **Appendix A.2**.

## 4.2 Implementation Details

We utilize pretrained encoders for feature extraction: CLIP [22]+SlowFast [5] or InternVideo2 [29] for QVHighlights [10] and Charades-STA [6], and I3D [2]+CLIP for TVSum [23]. Features are extracted without fine-tuning. Detailed feature extraction settings, full training setups and hyperparameters are described in the **Appendix A.4**.

## 4.3 Experiment Results

We evaluate our model on the QVHighlights [10] dataset, which supports both Moment Retrieval (MR) and Highlight Detection (HD). Test and validation results are shown in Table 1 and Table 2, respectively. Across both backbones—CLIP+SlowFast and InternVideo2—DualGround consistently surpasses prior methods [1, 13, 18, 24, 31] in MR metrics, especially at higher IoU thresholds (e.g., R1@0.7), highlighting its strength in precise moment localization. Table 2 ensures fair comparison by using InternVideo2 for all methods. Even under this strong backbone, DualGround achieves the best performance across all metrics, confirming that the improvements stem from our architecture, not just feature quality. Notably, we improve R1@0.7 by **1.72%** with CLIP+SlowFast and **2.98%** with InternVideo2.

In Table 5, we further evaluate on the Charades-STA [6] benchmark on CLIP and Internvideo2 backbone feature. DualGround again shows consistent gains in both R1@0.5 and R1@0.7, under both backbone settings. This reinforces the generalization ability of our model across different domains and video types.

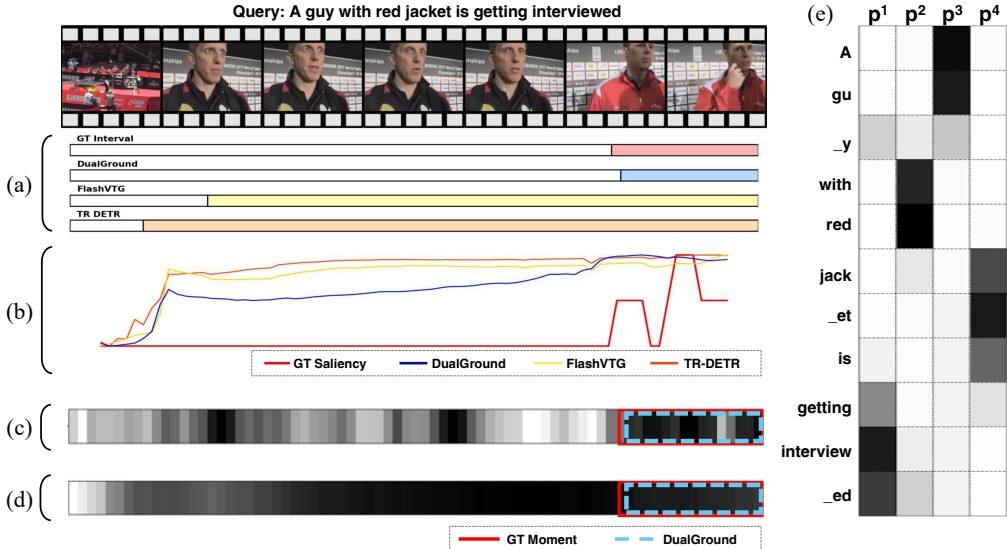

Figure 3: Visualization results on the QVHighlights validation split. (a) Moment retrieval predictions and (b) Highlight detection scores are compared across models. (c) L2 norm activation map of phrase-level embeddings, (d) L2 norm activation map of sentence-level embeddings, and (e) Phrase-to-word attention map are visualizations from our proposed DualGround model, highlighting how it captures localized semantics and structured alignment.

## 4.4 Qualitative Analysis

Figure 3 presents a qualitative result on the QVHighlights validation set, comparing our model with CG-DETR [18], TR-DETR [24], and FlashVTG [1]. In the visualization, baseline models relying heavily on sentence-level [EOS] representations tend to focus on the prolonged *interviewing* scene from the beginning of the video, underutilizing the local semantic cue of the phrase *"red jacket"*. In contrast, our model leverages phrase-level representations that preserve word-level semantics, allowing it to accurately localize the intended moment corresponding to the described visual concept.

In addition, the visualization highlights how our phrase grouping mechanism effectively clusters contextually related words into coherent units. These phrase clusters align well with localized visual cues, demonstrating the benefit of disentangled phrase representations for fine-grained temporal localization. Additional visualization results can be found in the **Appendix E.**

## 4.5 Ablation & Detailed Analysis

**Ablation Study.** Table 3 presents the results of our ablation study conducted on the QVHighlights validation split. Setting (a) corresponds to the baseline model FlashVTG [1], which employs a full-token sequence setting, including both word and the [EOS] token. All subsequent settings (b)–(g) are based on our proposed dual-path architecture, where the sentence-level and phrase-level branches are jointly optimized. We observe consistent performance gains in MR when replacing flat word-level modeling with structured phrase-level representations. Specifically, introducing phrase clustering significantly improves both R1@0.7 and mAP compared to directly modeling context between individual word tokens and video clips. Furthermore, incorporating the Distinct Query Attention loss ($L_{\text{DQA}}$), which encourages semantic separation across phrases, yields additional gains, demonstrating its effectiveness in enhancing phrase-level disentanglement and improving temporal localization.

While the improvements are clear for MR, we observe that HD exhibits more nuanced behavior. Certain settings without proper phrase regulation (e.g., when omitting $L_{\text{DQA}}$ or $L_{\text{EOS}}$) lead to slight degradations in HD metrics, suggesting that the unregulated phrase-level path can inject noisy fine-grained cues into the sentence-level representation. This highlights the importance of phrase

Table 3: Ablation study on QVHighlights-val split. RPG, $L_{\text{DQA}}$, and $L_{\text{EOS}}$ denote Recurrent Phrase Generation, Distinct Query Attention Loss, and `[EOS]` Reconstruction Loss, respectively.

| Settings | RPG | Slot | $L_{\text{DQA}}$ | $L_{\text{EOS}}$ | R1@0.7 | mAP | VG-Hit@1 | VG-mAP |
|---|---|---|---|---|---|---|---|---|
| (a) | | | | | 56.13 | 52.24 | 70.88 | 44.04 |
| (b) | | ✓ | ✓ | ✓ | 56.99 | 52.46 | 71.12 | 43.97 |
| (c) | ✓ | | ✓ | ✓ | 57.83 | 53.02 | 71.29 | 44.09 |
| (d) | ✓ | ✓ | | | 56.17 | 52.30 | 70.07 | 43.34 |
| (e) | ✓ | ✓ | ✓ | | 58.02 | 53.11 | 70.28 | 43.51 |
| (f) | ✓ | ✓ | | ✓ | 56.55 | 52.53 | **71.77** | 44.10 |
| (g) | ✓ | ✓ | ✓ | ✓ | **58.97** | **53.26** | 71.62 | **44.12** |

Table 4: MR mAP on QVHighlights val split by query length using IV2 backbone.

| Method | # Words | | | |
|---|---|---|---|---|
| | 0–10 | 10–15 | 15–20 | >20 |
| CG-DETR [18] | 48.61 | 50.88 | 47.80 | 35.21 |
| TR-DETR [24] | 49.26 | 49.15 | 50.26 | 35.33 |
| FlashVTG [1] | 53.24 | **54.78** | 50.58 | 43.46 |
| **DualGround** | **54.12** | 54.33 | 51.96 | 48.92 |

Table 5: MR Performance on Charades test set under different backbones.

| Method | Backbone | R1@0.5 | R1@0.7 |
|---|---|---|---|
| UniVTG [11] | C+SF | 59.25 | 36.64 |
| CG-DETR [18] | C+SF | 58.41 | 36.32 |
| TR-DETR [24] | C+SF | 57.61 | 33.52 |
| FlashVTG [1] | C+SF | 61.08 | 37.89 |
| **DualGround** | C+SF | **61.11** | **38.52** |
| CG-DETR [18] | IV2 | 70.40 | 48.40 |
| TR-DETR [24] | IV2 | 69.73 | 46.33 |
| FlashVTG [1] | IV2 | 70.32 | 49.87 |
| **DualGround** | IV2 | **70.67** | **50.33** |

regulation in preserving global semantic consistency during clip-level saliency modeling. Overall, the results confirm that our dual-path design enhances MR performance while maintaining comparable or improved HD performance across most settings.

**Across different query lengths.** We further analyze the impact of query length on performance, as shown in Table 4. While existing methods show a clear performance drop as queries become longer, our **DualGround** maintains robust performance even for queries exceeding 20 words. This suggests that our phrase-level path plays a critical role in handling complex queries where the global `[EOS]` token alone is insufficient to capture the full semantic structure.

## 5 Conclusion, Limitation, and Future Works

**Conclusion.** We propose DualGround, a dual-branch architecture for Video Temporal Grounding that separates sentence-level and phrase-level semantics. Unlike prior models relying on the sentence-level `[EOS]` token, ours introduces structured phrase modeling to recover fine-grained local cues. This enables richer video-text alignment by capturing both global intent and local context. We hope this work lays the groundwork for future research in semantic disentanglement and multimodal grounding.

**Limitations & Future Works.** Our model assumes a fixed number of phrases per query, requiring manual adjustment across datasets. Additionally, it does not leverage audio features, which may limit performance in audio-visual grounding scenarios. Adaptive phrase segmentation based on query structure and extension to audio signals for richer multimodal grounding are promising directions for future research.

## 6 Acknowledgement

This research was supported by the National Research Foundation of Korea (NRF) grant funded by the Korea government (MSIT) (No. RS-2024-00340745), and the Yonsei Signature Research Cluster Program of 2025 (2025-22-0013), and the the Korea Institute of Science and Technology (KIST) Institutional Program (Project No.2E33612-25-016).

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

## Appendix

The appendix is organized as follows.

- **Dataset, Metric, Implementation Details:** We describe the datasets used, including highlight detection results on TVSum, the evaluation metrics employed, and key implementation details.

- **Token Dependency Analysis:** We examine the model's reliance on the `[EOS]` token through attention correlation analysis, performing ablations under various token-conditioning settings with CLIP- and InternVideo2-based encoders.

- **Analysis of Phrase Segment:** We investigate how the optimal number of segmented phrases varies depending on the dataset and backbone features, analyzing its impact on model performance.

- **Ablation on Fusion Method:** We compare multiple strategies for fusing clip-level embeddings from the phrase and sentence paths.

- **Additional Visualization:** We include supplementary visualizations to better illustrate the model behavior and support claims made in the main paper.

## A  Dataset, Metric, Implementation Details

### A.1  Dataset Description

**QVHighlights**   QVHighlights [10] is a large-scale benchmark for joint video moment retrieval and highlight detection. It contains 10,148 videos collected from YouTube, spanning various domains including daily life, travel, and news. Each video is paired with natural language queries and annotated with corresponding highlight segments.

**Charades-STA**   Charades-STA [6] extends the Charades dataset by adding temporal moment annotations aligned with text queries. It consists of 9,848 short videos depicting indoor human activities and provides 16,128 annotated query-moment pairs. The dataset is commonly used for evaluating moment retrieval performance and is provided with a standard train/test split.

**TVSum**   TVSum [23] is a video summarization dataset comprising 50 videos from 10 different categories such as documentary, sports, and travel. Each video is annotated with frame-level importance scores gathered through crowd-sourced annotations. Following prior work, we adopt a 4:1 train-test split and use video titles as textual queries in the highlight detection setting. Although originally intended for summarization, TVSum is widely repurposed for highlight detection due to the similarity between the two tasks.

### A.2  Evaluation Metrics

We employ standard metrics commonly used in moment retrieval and highlight detection tasks. **Recall@1** is measured at multiple Intersection over Union (IoU) thresholds (e.g., 0.5 and 0.7), indicating whether the top-ranked prediction sufficiently overlaps with any ground-truth segment. **Mean Average Precision (mAP)** is computed by averaging the precision across multiple IoU thresholds, capturing both retrieval quality and temporal localization accuracy. **Hit@1** evaluates whether the top-scoring prediction exactly matches one of the ground-truth highlights, serving as a strict top-1 correctness measure. Additionally, **mean IoU (mIoU)** reports the average overlap between predicted and annotated segments.
We report Recall@1 (0.5/0.7), mAP, and Hit@1 on **QVHighlights**, Recall@1 (0.5/0.7) and mean IoU on **Charades-STA**, and top-5 mAP and Hit@1 on **TVSum**.

### A.3  Experiment Results on TVSum Dataset

Table A1 presents the highlight detection performance on the TVSum *val* split across 10 video categories. Our method achieves the highest average mAP of **88.1**, outperforming existing base-

Table A1: Experimental results on the TVSum *val* dataset.

| Method | VT | VU | GA | MS | PK | PR | FM | BK | BT | DS | Avg |
|---|---|---|---|---|---|---|---|---|---|---|---|
| LIM-S [32] | 55.9 | 42.9 | 61.2 | 54.0 | 60.4 | 47.5 | 43.2 | 66.3 | 69.1 | 62.6 | 56.3 |
| Trailer [28] | 61.3 | 54.6 | 65.7 | 60.8 | 59.1 | 70.1 | 58.2 | 64.7 | 65.6 | 68.1 | 62.8 |
| SL-Module [34] | 86.5 | 68.7 | 74.9 | 86.2 | 79.0 | 63.2 | 58.9 | 72.6 | 78.9 | 64.0 | 73.3 |
| UMT [14] | 87.5 | 81.5 | 88.2 | 78.8 | 81.4 | 87.0 | 76.0 | 86.9 | 84.4 | 79.6 | 83.1 |
| QD-DETR [19] | 88.2 | 87.4 | 85.6 | 85.0 | 85.8 | 86.9 | 76.4 | 91.3 | 89.2 | 73.7 | 85.0 |
| UVCom [31] | 87.6 | 91.6 | 91.4 | 86.7 | 86.9 | 86.9 | 76.9 | 92.3 | 87.4 | 75.6 | 86.3 |
| CG-DETR [18] | 86.9 | 88.8 | **94.8** | **87.7** | 86.7 | 89.6 | 74.8 | 93.3 | 89.2 | 75.9 | 86.8 |
| TR-DETR [24] | 89.3 | 93.0 | 94.3 | 85.1 | 88.0 | 88.6 | **80.4** | 91.3 | 89.5 | 81.6 | **88.1** |
| FlashVTG [1] | 88.3 | **94.3** | 91.5 | **87.7** | 87.1 | 91.1 | 74.7 | **93.4** | 90.3 | 81.7 | 88.0 |
| **DualGround** | **89.7** | 93.2 | 90.7 | 87.4 | **88.3** | **91.3** | 75.6 | 92.4 | **90.6** | 81.9 | **88.1** |

Table A2: Implementation details across datasets. From top to bottom, we list the hyperparameters and architectural configurations for QVHighlights (QVH.), Charades (Ch.), and TVSum (TVS.). In the **Feat** column, SF+C denotes the use of SlowFast and CLIP features, IV2 refers to InternVideo2, and I3D indicates I3D features. From left to right, **bs** is the batch size, **E** is the number of training epochs, and **lr** is the learning rate. **Ld** and **N** represent the counts of dummy tokens and phrase segments, respectively. **D.Enc** specifies the depth of dummy encoders, **ACA** is the number of adaptive cross-attention layers, and **P-SA** indicates the number of slot attention layers in the phrase-level path. **P.Enc** and **S.Enc** denote self-attention layers applied along the clip axis in the phrase-level and sentence-level paths, respectively. $\lambda_{MR}, \lambda_{HD}, \lambda_{phrase}$ are loss weights for moment retrieval, highlight detection, and phrase-level supervision. $r_{DQA}$ is a coefficient controlling the orthogonality regularization in the DQA loss.

| Dataset | Feat | Hyperparameter | | | | | Layer # | | | | | Loss | | | |
|---|---|---|---|---|---|---|---|---|---|---|---|---|---|---|---|
| | | bs | E | lr | Ld | N | D.Enc | ACA | P-SA | P.Enc | S.Enc | $\lambda_{MR}$ | $\lambda_{HD}$ | $\lambda_{phrase}$ | $r_{DQA}$ |
| QVH. | SF+C | 64 | 150 | $1e^{-4}$ | 3 | 4 | 2 | 3 | 2 | 2 | 2 | 5 | 1 | 1 | 0.3 |
| QVH. | IV2 | 64 | 150 | $1e^{-4}$ | 3 | 4 | 2 | 3 | 2 | 2 | 2 | 5 | 1 | 1 | 0.3 |
| Ch. | SF+C | 128 | 50 | $2.5e^{-4}$ | 3 | 3 | 2 | 3 | 2 | 2 | 2 | 5 | 1 | 1 | 0.3 |
| Ch. | IV2 | 128 | 50 | $2.5e^{-4}$ | 3 | 3 | 2 | 3 | 2 | 2 | 2 | 5 | 1 | 1 | 0.3 |
| TVS. | I3D | 4 | 600 | $1e^{-3}$ | 3 | 3 | 2 | 3 | 2 | 2 | 2 | 5 | 1 | 1 | 0.3 |

lines including TR-DETR and FlashVTG. Notably, our model exhibits strong consistency across Parade(**PR**), Attempting a Bike Trick(**BT**), and Dog Show(**DS**).

## A.4 Implementation Details

Table A2 summarizes the training configurations across datasets. We vary the backbone features (SF+C, IV2, I3D) depending on the dataset and adopt consistent architectural settings. Specific hyperparameters, layer numbers, and loss coefficients are detailed in the table.

Each model uses a hidden dimension of 256 and is optimized with the AdamW optimizer. Transformer layers follow a post-norm architecture with 8 attention heads. For post-processing, non-maximum suppression (NMS) is applied with a threshold of 0.7. All experiments are conducted on a machine equipped with a Ryzen 3960X 24-core CPU and a single NVIDIA RTX 3090 GPU.

For the **InternVideo2 (IV2)** [29] setting, we employ the pretrained model released by OpenGVLab. The video encoder corresponds to the 1B-parameter version of InternVideo2-stage2, while the text encoder is stage2-CLIP version (**InternVL-7B**) to enhance cross-modal representation quality. This configuration follows the official IV2–CLIP training pipeline and maintains consistent alignment between visual and textual embeddings.

## B  Token Dependency Analysis

We quantitatively evaluate the model's dependency on the [EOS] token by measuring correlations of cross-modal attention pattern across tokens, which reveal the degree of over-reliance by the [EOS]

Table A3: Performance of VTG models using the **SF+C** backbone across token conditions.

| Method | Word | EOS | Full | R1@0.5 | R1@0.7 | mAP | mAP@0.5 | mAP@0.75 |
|---|---|---|---|---|---|---|---|---|
| CG-DETR | ✓ | | | 64.84 | 49.68 | 43.18 | 65.27 | 44.16 |
| CG-DETR | | ✓ | | 62.19 | 46.13 | 41.87 | 64.01 | 42.43 |
| CG-DETR | | | ✓ | 66.90 | 50.32 | 43.47 | 65.48 | 44.79 |
| TR-DETR | ✓ | | | 66.32 | 50.45 | 43.99 | 65.74 | 44.89 |
| TR-DETR | | ✓ | | 64.00 | 47.74 | 41.78 | 64.25 | 42.45 |
| TR-DETR | | | ✓ | 66.48 | 50.71 | 44.53 | 65.43 | 44.98 |
| FlashVTG | ✓ | | | 68.85 | 53.81 | 48.42 | 67.83 | 51.50 |
| FlashVTG | | ✓ | | 65.62 | 52.60 | 45.32 | 67.12 | 50.49 |
| FlashVTG | | | ✓ | _69.03_ | 54.06 | _49.85_ | _68.44_ | _52.12_ |
| DualGround | ✓ | | | 68.20 | _54.11_ | 48.51 | 68.02 | 51.83 |
| DualGround | | ✓ | | 65.91 | 52.31 | 45.44 | 67.24 | 50.33 |
| DualGround | | | ✓ | **69.25** | **54.87** | **49.96** | **68.62** | **52.30** |

Table A4: Performance of VTG models using the **IV2** backbone across token conditions.

| Method | Word | EOS | Full | R1@0.5 | R1@0.7 | mAP | mAP@0.5 | mAP@0.75 |
|---|---|---|---|---|---|---|---|---|
| CG-DETR | ✓ | | | 70.06 | 55.55 | 48.84 | 69.71 | 49.66 |
| CG-DETR | | ✓ | | 71.35 | 56.65 | 49.36 | 70.08 | 50.67 |
| CG-DETR | | | ✓ | 69.74 | 56.45 | 48.97 | 69.18 | 50.46 |
| TR-DETR | ✓ | | | 70.65 | 55.94 | 48.80 | 69.52 | 49.57 |
| TR-DETR | | ✓ | | _73.35_ | _58.84_ | 50.19 | 72.02 | 52.20 |
| TR-DETR | | | ✓ | 72.06 | 57.03 | 49.23 | 70.45 | 50.83 |
| FlashVTG | ✓ | | | 70.72 | 55.90 | 51.33 | 70.92 | 52.80 |
| FlashVTG | | ✓ | | 72.23 | 56.51 | 52.19 | 72.34 | _55.60_ |
| FlashVTG | | | ✓ | 72.32 | 56.89 | _52.26_ | _72.39_ | 55.21 |
| DualGround | ✓ | | | 72.20 | 57.71 | 51.70 | 72.28 | 55.29 |
| DualGround | | ✓ | | 72.11 | 56.55 | 52.24 | 72.31 | 55.33 |
| DualGround | | | ✓ | **73.48** | **58.97** | **53.26** | **72.99** | **56.35** |

token. We then analyze how varying textual token conditions [Word only, [EOS] only, and Full (Word + [EOS]] affect the performance of VTG models under two backbone settings: CLIP-based (SF+C) and InternVideo2-based (IV2). Table A3 and Table A4 present the moment retrieval results on the QVHighlights *val* set across these conditions.

## B.1 Generalization of the EOS Over-Reliance

To verify that the over-reliance on the [EOS] token is prevalent phenomenon, we conduct a quantitative correlation analysis across the entire dataset. Specifically, we measure the statistical correlation between the attention weights assigned to the [EOS] token and those assigned to individual word tokens during cross-modal interaction. We adopt both the **Pearson** and **Spearman** correlation coefficients, which evaluate linear and rank-based relationships, respectively, to ensure robustness.

**Pearson Correlation.** Pearson correlation coefficient between two variables $x$ and $y$ is defined as:

$$r = \frac{\sum_{i=1}^{N}(x_i - \bar{x})(y_i - \bar{y})}{\sqrt{\sum_{i=1}^{N}(x_i - \bar{x})^2}\sqrt{\sum_{i=1}^{N}(y_i - \bar{y})^2}},$$

(7)

where $x_i$ and $y_i$ denote individual data points, and $\bar{x}$, $\bar{y}$ are their mean values. A higher $r$ indicates a stronger linear relationship between $x$ and $y$.

**Spearman Correlation.** Spearman rank correlation assesses monotonic relationships based on ranked values:

$$\rho = 1 - \frac{6\sum_{i=1}^{N} d_i^2}{N(N^2 - 1)}, \tag{8}$$

where $d_i$ is the rank difference between the paired values, and $N$ is the number of data points.

**Measurement Procedure.** The overall computation process is summarized as follows:

1. **Cross-modal attention extraction:** For each sample in the training and validation sets, we extract the cross-modal attention map $\mathbf{A} \in \mathbb{R}^{N_t \times N_v}$, where $N_t$ and $N_v$ denote the number of text tokens and video clips, respectively.

2. **Token isolation:** We separate the attention vector of the [EOS] token, $\mathbf{a}_{\text{EOS}}$, and those of the remaining $N_t - 1$ word tokens $\{\mathbf{a}_i\}_{i=1}^{N_t - 1}$.

3. **Token-wise correlation:** For each word token, we compute Pearson and Spearman correlations between $\mathbf{a}_i$ and $\mathbf{a}_{\text{EOS}}$, yielding $(N_t - 1)$ correlation values per sample.

4. **Averaging:** We average the correlations across tokens and then across all samples within the subset, reporting mean Pearson and Spearman values for both training and validation sets.

**Results.** Table A5 summarizes the results for representative VTG models.

Table A5: Average Pearson and Spearman correlations between [EOS] and word-token attentions.

| Model | Train | | Val | |
|---|---|---|---|---|
| | Pearson | Spearman | Pearson | Spearman |
| CG-DETR | 0.8960 | 0.8914 | 0.5962 | 0.7622 |
| TR-DETR | 0.8110 | 0.7753 | 0.6021 | 0.6340 |
| FlashVTG | 0.9745 | 0.9801 | 0.6771 | 0.7800 |

**Discussion.** Across all models, the correlation values remain consistently high (close to 1.0) for both Pearson and Spearman metrics, indicating that word tokens exhibit attention patterns highly similar to that of the [EOS] token. This confirms that prior VTG models show a generalized over-reliance on [EOS], where word-level semantics are largely overridden by global sentence-level alignment cues.

Furthermore, when relating these findings to the results in TableA4, we observe that models with weaker attention correlations tend to yield higher performance under the the single [EOS] token setting, when compared with Full-token setting. This suggests that in current model architectures, suppressing the local semantic contributions of individual word tokens may lead to a more optimized training trajectory.

### B.2 Impact of Backbone Semantics

Under the SF+C backbone (Table A3), using the [EOS] token alone consistently yields lower performance than using word tokens across all models. In contrast, the IV2 backbone (Table A4) shows the opposite trend: in all models except ours, the [EOS]-only setting achieves either the best performance (e.g., CG-DETR, TR-DETR) or results comparable to other configurations (e.g., FlashVTG).

We attribute this discrepancy to the difference in feature dimensionality between the backbones. CLIP encodes each token as a 512-dimensional vector, while InternVideo2 produces 4096-dimensional embeddings. This higher capacity allows IV2's [EOS] token to carry richer sentence-level semantics, enabling strong alignment even without word-level information.Conversely, CLIP's limited [EOS] capacity cannot fully represent complex queries, leading models to fall back on word tokens for localized cues. However, this reliance arises not from an intentional design but as a side effect of the [EOS] token's limitations. Treating all tokens uniformly in a flat sequence still ignores their distinct semantic roles, leading to suboptimal alignment.

Our proposed method alleviates this issue by separating sentence-level and phrase-level semantics. As shown in Table A3, it achieves robust performance even with CLIP-based features, validating its

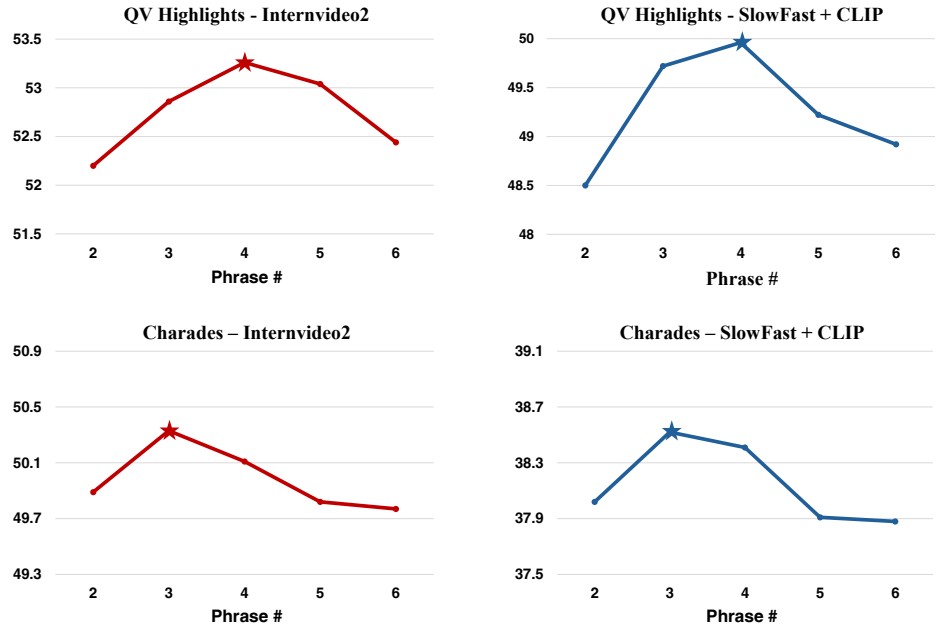

Figure A1: Ablation study on the number of phrase segments.

effectiveness despite the limited capacity of the [EOS] token. As vision-language models (VLMs) evolve with increasingly powerful text encoders, the [EOS] token will likely play an even greater role, making proper treatment of token-level semantics a critical consideration for future VTG architectures.

Table A6: Average query length per dataset.

| Dataset | Split | Query Length |
|---|---|---|
| QVHighlights | train | 10.46 |
| | val | 10.49 |
| Charades | train | 6.21 |
| | val | 6.23 |
| TVSum | train | 7.55 |
| | val | 7.70 |

Table A7: Ablation on Fusion Method

| Option | R1@0.5 | R1@0.7 | mAP |
|---|---|---|---|
| Add | 73.48 | **58.97** | **53.26** |
| Hadamard | 71.71 | 55.25 | 51.27 |
| Gate | 73.51 | 58.71 | 53.24 |
| Concat-mlp | **73.66** | 58.19 | 52.91 |

## B.3 Architectural Influence on Token Utilization

As shown in Table A4, CG-DETR and TR-DETR achieve better performance when using only the [EOS] token, compared to word or full token inputs. This suggests that word tokens may act as noise in these architectures. In CG-DETR, the clip-word distillation loss emphasizes alignment with individual words, which can suppress the rich global semantics of the [EOS] token. In TR-DETR, the global textual feature used for regulation is computed by mean-pooling over all word tokens. This strategy may dilute the semantic strength of the [EOS] token and introduce noise from weakly aligned or irrelevant words. In both cases, using only the [EOS] token avoids such noise and leads to better alignment.

These results suggest that the integration of token-level inputs should account for the distinct semantic roles of word and [EOS] tokens. Word tokens are most effective when they complement the global sentence representation without interfering with it. Our DualGround framework supports this balance by explicitly disentangling global and local semantics.

# C   Analysis of Phrase Segment

## C.1   Ablation on Phrase Segment Number

To determine the optimal number of phrase segments, we conduct an ablation study on the phrase segmentation parameter $N$, which defines the number of semantic units extracted from the input query. We experiment with different values of $N$ on both the Charades and QVHighlights datasets using two backbones: **SlowFast + CLIP** and **Internvideo2**.

As shown in Fig. A1, Charades achieves the best performance at $N = 3$, while QVHighlights yields the highest accuracy at $N = 4$. This difference is further analyzed in the next subsection. We also observe a performance drop when $N$ becomes large. Excessive segmentation divides queries into overly short spans, which may fail to capture complete semantic units and lead to fragmented or diluted phrase representations. This prevents effective alignment with video content and undermines the benefits of phrase-level modeling.

## C.2   Effect of Query Complexity

We hypothesize that the optimal number of phrase segments is influenced by the complexity of text queries. Intuitively, queries with greater semantic richness benefit from finer phrase decomposition, as they contain more diverse word-level information that can be aligned with visual content.

To investigate this, we analyze the average query length across datasets, as shown in Tab. A6. Queries from QVHighlights are substantially longer than those from Charades or TVSum, indicating higher semantic complexity. This aligns with our ablation results, where QVHighlights achieves the best performance at $N = 4$, while Charades performs best at $N = 3$. These observations suggest that phrase segmentation should be tailored to the dataset's linguistic characteristics.

# D   Ablation on Fusion Strategy

We evaluate four different strategies for integrating the sentence-level ($V_s$) and phrase-level ($V_p$) features into a unified representation $F = V_s + V_p$, which is used for downstream prediction (see Sec. 3.4). The following options are compared in Tab. A7:

- **Add:** Element-wise addition of $V_s$ and $V_p$. This is our default configuration due to its simplicity and efficiency.
- **Hadamard:** Element-wise multiplication of $V_s$ and $V_p$, emphasizing shared dimensions.
- **Gate:** A learnable sigmoid gate $\sigma$ is applied such that the fused feature $F = \sigma \cdot V_s + (1 - \sigma) \cdot V_p$. This allows the model to adaptively weight sentence and phrase contributions.
- **Concat-mlp:** The two features are concatenated and passed through a linear projection layer to match the original dimensionality.

As shown in Tab. A7, the **Add** method achieves the best overall performance considering both effectiveness and computational simplicity. While *Concat-mlp* slightly improves R1@0.5, its performance on R1@0.7 and mAP is inferior to *Add*. The *Gate* mechanism performs comparably but introduces additional parameters and complexity. We thus adopt **addition** as our default fusion strategy due to its favorable trade-off between accuracy and efficiency.

# E   Additional Visualization

We provide additional qualitative results in Figure A2. The visualizations demonstrate how semantically aligned word tokens are clustered into meaningful phrases, as illustrated in A2(e). This grouping provides localized, clip-wise information that complements the global sentence-level representation, particularly in cases where fine-grained cues are difficult to capture. As a result, it enables more accurate and context-aware temporal grounding.

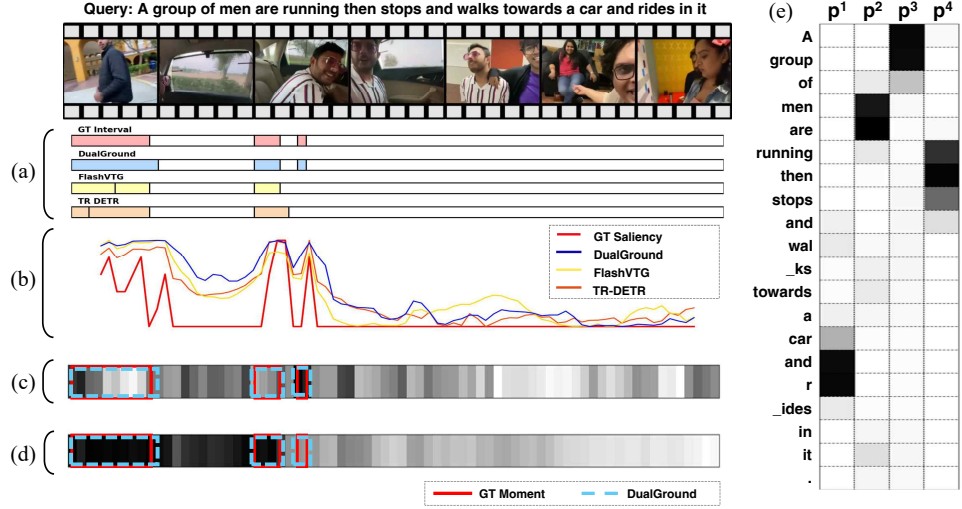

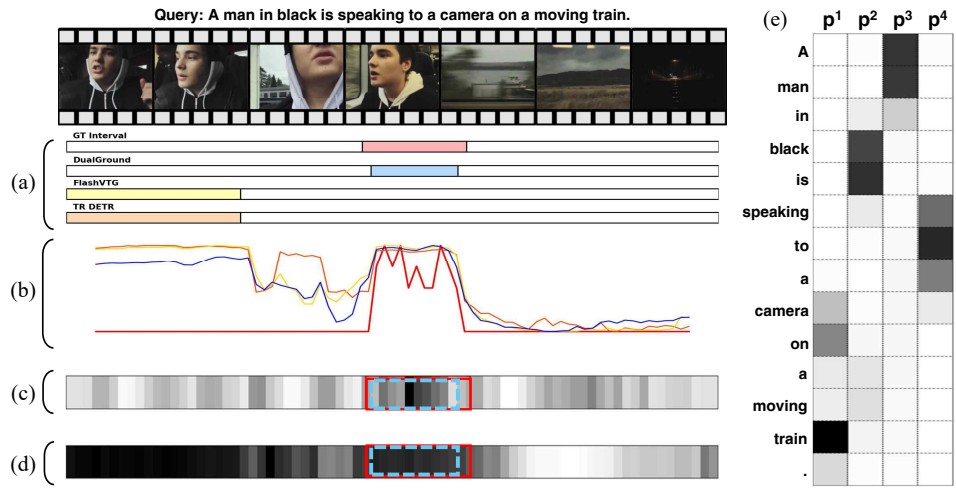

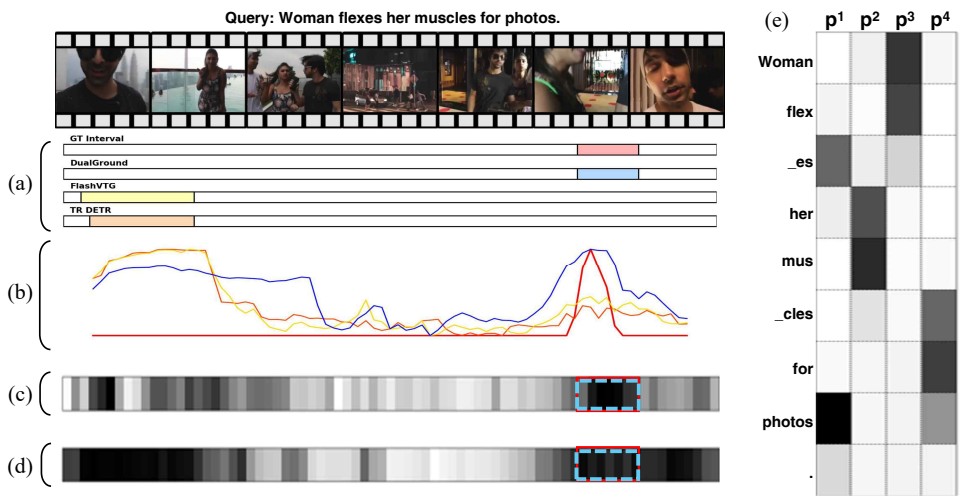

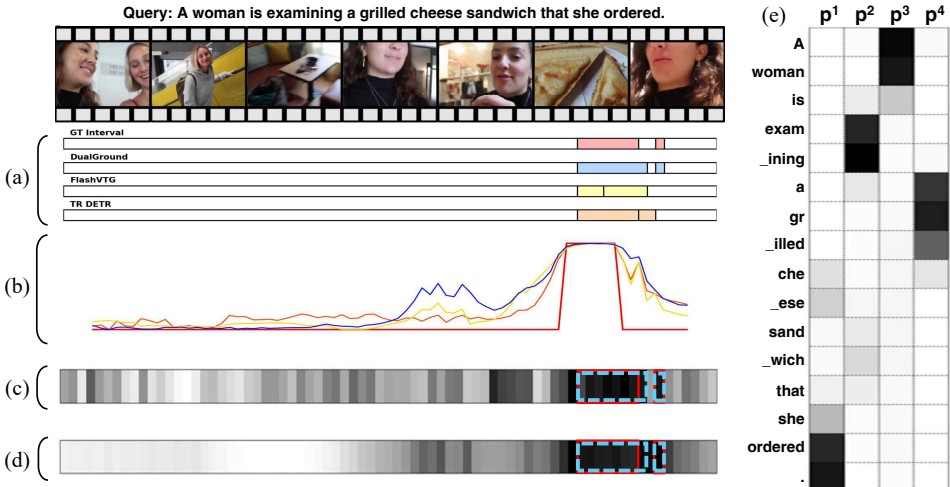

Figure A2: Additional Visualization results on the QVHighlights validation split. (a) Moment retrieval predictions and (b) Highlight detection scores are compared across models. (c) L2 norm activation map of phrase-level embeddings, (d) L2 norm activation map of sentence-level embeddings, and (e) Phrase-to-word attention map are visualizations from our proposed DualGround model, highlighting how it captures localized semantics and structured alignment.

