# OpenReview forum: "Empower Words: DualGround for Structured Phrase and Sentence-Level Temporal Grounding"
_NeurIPS.cc/2025/Conference — NeurIPS 2025 poster_

### Official Review · Reviewer_3BT3 · 2025-06-22

**Clarity:** 2
**Significance:** 3
**Originality:** 3
**Rating:** 4
**Confidence:** 3

**Summary:**

This paper proposes DualGround, a dual-branch architecture for Video Temporal Grounding (VTG), which aims to localize relevant moments in untrimmed videos given a natural language query. Existing VTG models often over-rely on the [EOS] token for global semantics, overlooking fine-grained word-level cues and limiting temporal precision. To address this, DualGround combines a sentence-level branch that leverages the [EOS] token with Adaptive Cross Attention, and a phrase-level branch that clusters and refines word tokens into structured phrases for localized grounding. Experiments on QVHighlights and Charades-STA demonstrate that DualGround achieves state-of-the-art performance on both Moment Retrieval and Highlight Detection tasks, showing enhanced robustness to complex queries.

**Questions:**

While the proposed dual-branch design is motivated by semantic disentanglement, the phrase-level branch is still trained to reconstruct the [EOS] embedding. This raises concerns about potential semantic redundancy between the two branches.

**Ethical Concerns:**

["NO or VERY MINOR ethics concerns only"]

**Final Justification:**

The authors have provided an explanation of the complementarity between the two existing branches and offered a comparison in terms of model parameters. The explanation regarding a fixed number of phrases is also reasonable.

**Limitations:**

yes

**Paper Formatting Concerns:**

No major formatting issues noticed.

**Quality:**

2

**Strengths And Weaknesses:**

Strengths:
1. The proposed DualGround introduces a clear and intuitive dual-branch architecture that separates global sentence-level and local phrase-level semantics, addressing a key blind spot in prior VTG models.
2. The paper convincingly demonstrates that existing methods over-rely on the [EOS] token through controlled ablation experiments and attention map analysis.
3. DualGround achieves state-of-the-art performance on both QVHighlights and Charades-STA benchmarks across multiple VTG metrics (e.g., R1@0.7, mAP, VG-Hit@1).
Weaknesses:
1. The model relies on a fixed number of phrases per query, introducing a dataset-specific hyperparameter that may hinder generalization and require manual tuning across different datasets and scenarios.
2. While the architecture separates global and local semantics, potential semantic redundancy between the sentence-level and phrase-level representations is not examined. The paper lacks analysis on whether the two branches contribute complementary information or overlap in functionality.
3. DualGround adds considerable architectural complexity—including dual semantic branches, Slot Attention, Recurrent Phrase Generation, and multiple loss functions—yet the performance gains over strong baselines like FlashVTG and TR-DETR are relatively modest (typically 1–2% mAP). Moreover, the paper does not provide any discussion of runtime cost, model size, or inference efficiency.

---

> ### Author Rebuttal · Authors · 2025-07-31
>
> # [W1] Fixed Number of Phrases
>
> As noted in our paper’s Limitation section, using a fixed number of phrase slots N across all queries is a constraint of our current design. Figure A1 in the supplementary material shows that the optimal N differs by dataset, reflecting variations in query length and linguistic complexity. This supports the reviewer’s suggestion that a dynamic strategy may offer better adaptability.
>
> Using a stable setting of **N=4**, our method consistently outperforms all baselines across both datasets, demonstrating robust and generalizable performance. For example, N=4 yields the best results on QVHighlights and causes only marginal drops on Charades (0.22 with InternVideo2, 0.09 with SF+C in R1@0.7).
>
> We also explored **dynamic clustering algorithms** such as FINCH, which operate directly on word tokens. However, these methods often struggled to form stable groupings, leading to optimization instability and inferior performance compared to our RPG-based formulation.
>
> A promising direction for future work is a hybrid approach that **sets a maximum number of clusters and dynamically allocates them based on query complexity**. While such designs may increase computational cost and pose challenges in defining and supervising query complexity, we believe they offer strong potential for further improving adaptability and performance across datasets.
>
> ***
> # [W2, Q1] Complementarity of Dual Branches
>
> We clarify that the two branches are designed to capture complementary semantics, not redundant information.
>
> Empirical evidence for the functional distinction between phrase and sentence branches is provided in visualization results ㅡL2 norm activation maps shown in Figure 3(c) (phrase-level) and Figure 3(d) (sentence-level) of the main paper, as well as Figure A2 in the supplementary material. The activation values indicate the degree of alignment between each pathway’s representation and the input video clips. These visualizations reveal that the two branches exhibit distinct activation patterns across time.
>
> To quantify distinction, we compute the correlation of clip-wise activation between phrase and sentence branches. The results, detailed in our responses to **Reviewer sNK1 [W3] and [W4]**, confirm that the two paths operate in complementary fashion.
>
> Moreover, EOS reconstruction in phrase path is not performed via direct supervision to replicate the [EOS] token embedding. Instead, contrastive InfoNCE loss is adopted, where the [EOS] embedding is treated as a positive target and contrasted against other samples in the batch. This design encourages the phrase-level representation to capture semantically aligned information sufficient to distinguish the correct [EOS], but it does not enforce exact replication of the [EOS]. As such, the phrase branch retains its own fine-grained compositional structure while remaining semantically coherent.
> Together, these findings support our claim that DualGround performs structured dual-path reasoning, rather than inducing semantic redundancy.
> ***
> # [W3] Parameter Observation
> We provide a quantitative comparison of the parameter size, FLOPs, and inference throughput in the table below:
> All measurements were conducted on a single NVIDIA RTX 3090 GPU, using a text token extracted by InternVideo2 (video feature length 768, text feature length 4096). For baselines (CG-DETR, TR-DETR, FlashVTG), we used their official base configurations including encoder/decoder layer. DualGround’s model setting follows the QVHighlights dataset configuration described in Supplementary Table A2.
>
> | Model | Params(M) | GFLOPs | Thorughput (samples/sec) | MR (mAP) | HD (VG-mAP) |
> |:---:|:---:|:---:|:---:|:---:|:---:|
> | **CG-DETR** | 942 | 0.48 | 78 | 48.93 | 42.30 |
> | **TR-DETR** | 737 | 0.40 | 116 | 48.93 | 43.74 |
> | **FlashVTG** | 754 | 0.46 | 103 | 52.61 | 44.08 |
> | **DualGround** | 883 | 0.51 | 84 | **53.26** | **44.12** |
>
> Although DualGround shows the highest FLOPs (0.51G), the increase over FlashVTG is only +0.05G, and inference remains real-time capable at 84 samples/sec. We argue that this slight increase is a reasonable trade-off, justified by the consistent performance gains in both Moment Retrieval and Highlight Detection.
> While the phrase-level path introduces additional computation, we design the architecture to offset this by reducing cost elsewhere:
>
> - Unlike prior models that apply cross-modal attention to full text tokens, DualGround routes only a single [EOS] token into the sentence-level path, effectively lightening the attention computation in that branch.
> - Both CG-DETR and FlashVTG utilize 40+ dummy tokens (45 and 40, respectively) to enhance representation richness across all token positions. In contrast, our method introduces only 3 dummy tokens (Ld=3), owing to the EOS-centric formulation of the sentence path, further reducing memory and compute.
> - At the word level, we perform lightweight clustering (e.g., N=4) via the Recurrent Phrase Generator, which processes (L-1) word tokens into N phrases. This drastically reduces the number of semantic units involved in phrase-level operations and ensures that the cost scales with N, not (L-1).
>
> Furthermore, the most time and computation consuming components in VTG pipelines are typically the pretrained vision-language encoders (e.g., InternVideo2, CLIP). In this context, the modest increase in FLOPs and parameter count introduced by our dual-branch architecture has negligible impact on the overall end-to-end inference latency.
> Therefore, we believe that the computational trade-off is practically acceptable.

---

> > ### Comment · Reviewer_3BT3 · 2025-08-05
> >
> > Thank you very much for the author's response. The authors have addressed my concerns, and as a result, I will increase my score.

---

> > > ### Author Response · Authors · 2025-08-05
> > >
> > > Thank you for your thoughtful consideration of our rebuttal and for updating your score accordingly. We truly appreciate your time and support.

---

### Official Review · Reviewer_ffuB · 2025-06-26

**Clarity:** 3
**Significance:** 3
**Originality:** 2
**Rating:** 5
**Confidence:** 4

**Summary:**

This paper addresses the task of Video Temporal Grounding (VTG), which involves localizing video segments corresponding to a natural language query. The authors begin by identifying a fundamental limitation in prior work: existing models treat all text tokens uniformly, leading to an over-reliance on the global semantics encoded in the special [EOS] token while under-utilizing more granular, word-level information essential for precise alignment. To solve this, the paper proposes DualGround, a novel dual-branch architecture that explicitly disentangles global and local semantics. One branch, the "sentence-level path," isolates the [EOS] token to handle coarse, global alignment. The other, more innovative "phrase-level path," clusters word tokens into semantically coherent phrases and models their fine-grained interactions with video clips. This path uses a Recurrent Phrase Generator and a Slot Attention module to create structured, context-aware phrase representations. By fusing the outputs of these two paths, DualGround achieves a more robust and precise video-text alignment. The method demonstrates state-of-the-art performance on standard VTG benchmarks, including QVHighlights and Charades-STA, validating the effectiveness of its disentangled semantic modeling approach.

**Questions:**

**1. On Comparison with Relevant Keyword-Centric Approaches**

The core idea of the phrase-level path is to identify and ground semantically important word groups. This concept shares similarities with recently proposed methods like 'Keyword-DETR' [1](Um et al., AAAI 2025), which also focuses on identifying key textual elements for grounding. Could the authors comment on the relationship between DualGround's phrase-level path and Keyword-DETR? A discussion of the key similarities and differences in the Related Work section, and potentially a comparative analysis, would help to better situate the novelty of your approach.

[1] Um, S. J., et al. "Watch Video, Catch Keyword: Context-aware Keyword Attention for Moment Retrieval and Highlight Detection." AAAI 2025.

**2. Ablation Study Results for Highlight Detection**

Table 3 provides a clear ablation study on the contributions of the proposed components for the Moment Retrieval (MR) task. However, the corresponding ablation results for the Highlight Detection (HD) task are not presented. Given that the model is designed for joint MR and HD, could the authors provide the ablation results for HD as well? This would be crucial to confirm that the proposed components are beneficial for both sub-tasks.

**3. Robustness of the EOS Reconstruction Loss**

The EOS Reconstruction Loss uses an InfoNCE objective, which works by treating other samples in the batch as negatives. This assumes that sentences from different video-query pairs are semantically distinct. However, in datasets that may contain many paraphrases or visually similar events described with slightly different sentences, this objective might incorrectly push apart the representations of semantically similar queries. Could the authors comment on the robustness of this loss component to datasets with high semantic similarity between different queries?

**4. On the Quantitative Analysis of Computational Cost**

The proposed phrase-level path, with its RPG and Slot Attention modules, introduces additional complexity compared to single-path architectures. While the paper demonstrates superior performance, a quantitative analysis of the computational cost is not provided. Could the authors provide details on the model's efficiency, for instance, by reporting metrics such as FLOPs and inference latency, and comparing them against key baselines like FlashVTG and TR-DETR? This information would be crucial for understanding the practical trade-offs of the `DualGround` architecture.

**Ethical Concerns:**

["NO or VERY MINOR ethics concerns only"]

**Final Justification:**

I thank the authors for their thorough and convincing rebuttal, which has successfully addressed my primary concerns. The new quantitative analysis of computational cost provides a clear picture of the practical trade-offs of the proposed architecture, and the additional ablation study for the Highlight Detection task confirms the benefits of the model's components for both sub-tasks. The authors have also provided a thoughtful comparison with related work and have agreed to include a discussion on societal impact. Given that my main reservations have been resolved, I believe the paper's strengths in its motivation and novel architecture are significant. Therefore, I have raised my score from Borderline Accept to Accept.

**Limitations:**

Yes. The authors have explicitly and clearly addressed the limitations of their work in the final "Conclusion, Limitation, and Future Works" section. They correctly identify the fixed number of phrases per query and the lack of audio features as key limitations and suggest them as promising directions for future research. This is excellent practice.

**Paper Formatting Concerns:**

.

**Quality:**

3

**Strengths And Weaknesses:**

**Strengths:**

- **Excellent Motivation and Problem Formulation:** The paper is exceptionally well-motivated. It starts with a clear and insightful empirical analysis (Figure 1) that convincingly demonstrates a core weakness in previous state-of-the-art models—their over-reliance on the [EOS] token. This strong diagnostic work makes the subsequent proposal of the `DualGround` framework highly compelling and easy to appreciate.
- **Novel and Elegant Architecture:** The proposed dual-branch architecture is a novel and elegant solution to the identified problem. The core idea of disentangling sentence-level (global) and phrase-level (local) semantics is powerful. The phrase-level path is particularly sophisticated, combining a Recurrent Phrase Generator (RPG) for initial grouping and a Slot Attention module for refinement. This demonstrates a deep understanding of structured representation learning and applies it effectively to the VTG task.
- **Thorough and Convincing Evaluation:** The experimental results are state-of-the-art and robustly validated. `DualGround` outperforms strong baselines across multiple standard benchmarks (QVHighlights, Charades-STA) and with different vision backbones. The comprehensive ablation studies (Table 3) clearly demonstrate the contribution of each novel component, and the analysis on varying query lengths (Table 4) provides strong evidence that the phrase-level path is key to the model's robustness on complex queries.

**Weaknesses:**

- **Architectural Complexity:** The phrase-level path, with its Recurrent Phrase Generator, Slot Attention module, and phrase-guided aggregation, is quite complex. While shown to be effective, this complexity might present implementation challenges and increase computational overhead. The paper could benefit from an analysis of the trade-offs between this complexity and performance.
- **Lack of Discussion on Negative Societal Impact:** In the NeurIPS checklist, the authors state that the work is foundational research without direct deployment risks and thus omit a discussion on negative societal impacts. While the direct risks may be low, VTG technology could be a component in large-scale content monitoring or surveillance systems. A brief acknowledgment of this common dual-use concern is generally good practice in the field.

---

> ### Author Rebuttal · Authors · 2025-07-31
>
> # [W1,Q4] Parameter Analysis
> We appreciate the reviewers’ concerns regarding the computational overhead introduced by our phrase-level architecture.
> To address this, we provide a quantitative comparison of model complexity and runtime efficiency against prior state-of-the-art baselines. The table below summarizes parameter counts, FLOPs, and inference throughput:
>
> | Model | Params(M) | GFLOPs | Thorughput (samples/sec) | MR (mAP) | HD (VG-mAP) |
> |:---:|:---:|:---:|:---:|:---:|:---:|
> | **CG-DETR** | 942 | 0.48 | 78 | 48.93 | 42.30 |
> | **TR-DETR** | 737 | 0.40 | 116 | 48.93 | 43.74 |
> | **FlashVTG** | 754 | 0.46 | 103 | 52.61 | 44.08 |
> | **DualGround** | 883 | 0.51 | 84 | **53.26** | **44.12** |
>
> All measurements were conducted on a single NVIDIA RTX 3090 GPU, using a text token extracted via InternVideo2 (video feature length 768, text feature length 4096). For baselines (CG-DETR, TR-DETR, FlashVTG), we used their official base configurations including encoder and decoder layers. DualGround’s configuration follows Supplementary Table A2.
>
> While DualGround has the highest FLOPs (0.51G), the increase over FlashVTG is only +0.05G, and the model still achieves real-time inference at 84 samples/sec. Given the consistent performance gains across Moment Retrieval and Highlight Detection, we believe this trade-off is justified. In addition, we carefully designed the architecture to offset complexity where possible:
>
> - **Sentence Path Efficiency:** Unlike previous works that apply cross-modal attention across all text tokens, DualGround passes only a single [EOS] token into the sentence path. This design significantly reduces attention computation in this branch.
> - **Fewer Dummy Tokens:** CG-DETR and FlashVTG use 45 and 40 dummy tokens, respectively, to maintain representational richness. In contrast, our EOS-centric sentence formulation requires only 3 dummy tokens, further reducing memory and compute cost.
> - **Efficient Phrase Operations:** The Recurrent Phrase Generator compresses (L-1) word tokens into N=4 phrase units, lowering the number of semantic elements involved in phrase-level operations. All downstream computations scale with N rather than (L-1), which ensures controlled overhead.
>
> Lastly, we note that the primary bottleneck in VTG pipelines typically lies in the pretrained VLM encoders (e.g., InternVideo2, CLIP). The incremental cost introduced by our dual-branch structure is marginal in the context of end-to-end latency and does not affect practical deployment.
> We believe DualGround maintains a favorable balance between architectural complexity and performance benefit. Our design provides interpretability and robustness with minimal additional cost, making it both effective and practically viable.
>
> ***
> # [W2] Negative Societal Impact
>
> We thank the reviewer for raising this point. While our work focuses on foundational modeling and is not directly intended for deployment, we acknowledge that Video Temporal Grounding (VTG) technology may be used as a component in broader systems with potential surveillance or content monitoring applications. We agree that it is good practice to recognize such negative societal impact, and we will revise our NeurIPS checklist to include a brief note on the potential societal implications and risks associated with downstream use.
> ***
> # [Q1] Comparison with Keyword-DETR
>
> Keyword-DETR [1] emphasizes visually sparse individual word tokens as key elements for alignment. The underlying intuition is that words referring to visually rare or distinctive concepts are more likely to serve as effective grounding cues. While this approach successfully highlights semantically informative words, it operates to weighting individual token, where alignment targets may be ambiguous— isolated word tokens such as “man” or “gray” can appear in a wide range of contexts, leading to imprecise grounding. This issue is amplified by tokenization, where a single meaningful word may span multiple subword tokens, further complicating visual alignment.
>
> Our approach is instead motivated by the observation that visually grounded concepts often correspond to multi-word expressions that convey a unified semantic meaning. Rather than assigning importance to individual word tokens, we focus on modeling interactions by grouping semantically related word tokens in the feature space. This approach captures compositional semantics more effectively and allows the model to attend to cohesive visual-linguistic patterns. We believe this design better supports the nature of semantic alignment required in moment retrieval tasks.
>
> In summary, while both approaches share a common goal of identifying linguistically salient cues for temporal grounding, they differ in their granularity and modeling strategies.
>
> - **Key Similarities**:
>   - **Salient textual cues**: Keyword-DETR and our approach aim to identify semantically significant cues in the text for grounding. Both methods recognize the importance of focusing on certain words or phrases that play a pivotal role in connecting visual content with textual cues.
>
>
> - **Key Differences**:
>   - **Granularity**: Keyword-DETR isolates individual word tokens, especially those that refer to visually rare or distinctive concepts. Our approach focuses on grouping semantically related word tokens into compositional phrases, which we believe captures compositional semantics more effectively.
>   - **Modeling Strategy**: Our model integrates **sentence-level interpretability** with **fine-grained token-level analysis**, allowing it to both capture broad semantic meanings and effectively handle detailed, context-dependent cues in visual grounding.
>
> [1] Um, Sung Jin, et al. "Watch Video, Catch Keyword: Context-aware Keyword Attention for Moment Retrieval and Highlight Detection." Proceedings of the AAAI Conference on Artificial Intelligence. Vol. 39. No. 7. 2025.
> ***
> # [Q2] Ablation Study on Highlight Detecion
> | Settings | RPG | Slot | L_DQA | L_EOS | MR-R1@0.7 | MR-mAP | HD-VG mAP | HD- VG Hit@1 |
> |:---:|:---:|:---:|:---:|:---:|:---:|:---:|:---:|:---:|
> | (a) |  |  |  |  | 56.13 | 52.24 | 44.04 | 70.88 |
> | (b) |  | ✔ | ✔ | ✔ | 56.99 | 52.46 | 43.97 | 71.12 |
> | (c) | ✔ |  | ✔ | ✔ | 57.83 | 53.02 | 44.09 | 71.29 |
> | (d) | ✔ | ✔ |  |  | 56.17 | 52.30 | 43.34 | 70.07 |
> | (e) | ✔ | ✔ | ✔ |  | 58.02 | 53.11 | 43.51 | 70.28 |
> | (f) | ✔ | ✔ |  | ✔ | 56.55 | 52.53 | 44.10 | 71.77 |
> | (g) | ✔ | ✔ | ✔ | ✔ | **58.97** | **53.26** | **44.12** | **71.62** |
>
> We have added HD evaluation metrics (VG-mAP and VG-Hit@1) to our existing ablation table to assess the effectiveness of each proposed component across both MR and HD tasks. This addition confirms that the proposed components contribute positively to both MR and HD tasks, validating the effectiveness of our dual-path design across settings. While MR aims to retrieve a discrete moment that best matches the sentence, HD requires assigning a saliency score to every clip based on the global sentence meaning.
>
> Notably, we observe that using the phrase-level path without phrase regulation ($L_{phrase}$, including $L_{DQA}$ and $L_{EOS}$) results in degraded HD performance. This degradation is particularly evident in the $L_{EOS}$ metric, which reflects the phrase path’s ability to preserve sentence-level semantics. Removing $L_{EOS}$ introduces noise into the sentence-level path by injecting unregulated fine-grained features. This highlights the importance of phrase regulation in preserving global semantic consistency during clip-level saliency modeling.
>
> While our proposed components yield consistent performance gains in both MR and HD, the improvement on HD metrics is comparatively modest. We attribute this to the partial misalignment between fine-grained local cues captured by the phrase-level path and the global saliency requirements of HD. The phrase-level path helps discover fine-grained moments, but these do not always aggregate effectively into global saliency estimates. We believe this limitation can be addressed by introducing adaptive mechanisms (e.g., gating) that dynamically balance global and local semantics. We will incorporate these HD ablation results and the above discussion into the revised manuscript.
> ***
> # [Q3] Robustness of the EOS Reconstruction Loss
>
> We appreciate the reviewer’s thoughtful question. We acknowledge that InfoNCE loss relies on the assumption that query sentences within a batch are semantically distinct, and that in datasets with frequent paraphrastic overlap, this may lead to false negatives, (i.e. semantically similar queries being treated as negatives) thereby incorrectly pushing their representations apart.
> In our current training setup, this issue is mitigated in practice by the diversity of sentence queries, which are derived from natural video descriptions that tend to vary across samples. Nonetheless, we recognize that in more paraphrase-rich datasets, this assumption may not always hold.
> To improve robustness in such cases, a practical extension involves leveraging a pretrained sentence encoder to compute semantic similarity between queries, thereby excluding highly similar pairs from the negative set. We believe this simple yet effective refinement can further enhance the stability and generalization of the InfoNCE objective.

---

> > ### Comment · Reviewer_ffuB · 2025-08-07
> >
> > Thank you for the detailed rebuttal. My concerns have been addressed, and I will raise my score accordingly.

---

### Official Review · Reviewer_sNK1 · 2025-07-01

**Clarity:** 3
**Significance:** 2
**Originality:** 3
**Rating:** 5
**Confidence:** 4

**Summary:**

The paper first highlights a characteristic of existing video grounding methods whereby they focus more on the EOS token representing the full sentence, rather than focusing on the fine-grained details of the sentence. The method, DualGround, aims to improve on previous Video Moment Retrieval and Highlight Detection approaches by implementing a phrase-level path alongside a sentence-level path. The phrase-level path generates phrase-level representations for interaction with the video features, replacing joint word-level and EOS token interaction. This phrase-level path seems to be the key novelty of the approach. The method achieves improved performance over state-of-the-art on Moment Retrieval and also strong performance on Highlight Detection.

**Questions:**

- In the supplementary, Table A3 and A4, is it understood why EOS alone is better in many cases with IV2 features, but with SF+C features, it is consistently worse with just EOS? This suggests that there is a fundamental difference in the features that is causing this effect rather than it being simply a case of the models not utilising the word tokens. As the claim that these models are not using the word tokens is a key part of the motivation of the method, it would be good to have a better understanding of this behaviour, as it does not seem to be true for the SF+C features, with which many of these models were originally designed (at least CG-DETR and TR-DETR were, and these are the ones that show the biggest discrepancy with IV2 features)
- It's not clear to me that using a phrase-level path is better than using a word-level path alongside a sentence-level path. So I am wondering if a similar improvement could be achieved in this manner using a word-level path. I understand this is likely not feasible in the rebuttal period but I wonder if any work has been done by the authors previously in this regard? If not, it could be interesting to test in the future (I'm not asking for it in the rebuttal period if it hasn't already been done). It would demonstrate further the effectiveness of the approach.
- Overall  my rating is positive as I think the method is logical and it shows generally strong performance, even if the performance gains are not extremely significant. I do, however, still believe that the motivation of the paper could be improved, hence keeping the rating at only borderline accept.

**Ethical Concerns:**

["NO or VERY MINOR ethics concerns only"]

**Final Justification:**

The rebuttal has addressed my main concerns. The motivation for the phrase-level path is maybe not quite as clear as I think it could be, although it still broadly gets the point across. Overall I think the phrase-level path is shown to be useful when the sentence-level representation does not fully capture the semantics of the query, the method produces strong enough results, and the method makes sense within the context of the use of the phrase-level path. I adjust my rating to accept.

**Limitations:**

Yes

**Quality:**

3

**Strengths And Weaknesses:**

Strengths
- The base-level motivation of the paper seems good, where it highlights that existing models treat word tokens and EOS tokens the same
- The idea of using phrase-level information seems sensible, and the manner in which this is implemented seems well thought through and makes sense.
- The performance is strong on Moment Retrieval, achieving reasonable, even if not extremely strong, performance gains, and performance is also competitive on Highlight Detection.
- The ablations are insightful, Table 3 shows the importance of each component and displays that the different components and losses work well in tandem with each other
- Table 4 is interesting, showing the particularly improved performance on longer sentence queries.
- It is appreciated that performance is tested with both SF+CLIP and IV2 features.
- The paper is generally well-written and the ideas are clearly described.

Weaknesses
- While the method appears to work well, achieving some performance gains, and the methodology largely makes sense, the main weaknesses that I see are related to the analysis of existing methods and the understanding of the motivation.
- I worry that Figure 1(b) could be somewhat misleading. In my estimation, it would not be that surprising or alarming that a video grounding model would pay more attention to the overall sentence representation than it would to the individual words. It would be more surprising to me if the model paid more attention to an individual word. Rather than comparing the attention of individual words, I would be more interested to see the comparison between the EOS token attention vs the summation of the attention from the other words. If the model was paying more attention to the EOS token than all of the word tokens combined that would be more of a clear issue and sign of bias in my mind.
- It would also be helpful if this could be summarised across a number of examples (ideally averaged across all examples) rather than just a single example. A single example does not provide much evidence for this behaviour. I think it would be beneficial for the paper if this motivation was more clear.
- Following from this, it would be nice to see something similar to Figure 1(b) for the DualGround model, to show that the model is actually paying more attention to the phrase features. Seeing as this is part of the motivation for the approach, it would be helpful to see that the method works to alter that aspect of the model behaviour. Admittedly, I'm not sure if there's a direct equivalent possible in the DualGround model due to the manner in which phrase information is incorporated into the video features. Perhaps the attention weights in the phrase-guided aggregation. If something like this was possible it would be nice to see in the supplementary.
- I think it's possible to say that it is debatable whether the model really achieves improved performance on Highlight Detection, as on QVHighlights test, FlashVTG does marginally better with IV2 features, while DualGround does marginally better with SF+C features. On QVHighlights val, DualGround does marginally better with IV2 features.
- Minor point: It would be helpful to associate f_enc in the text with the "dummy encoder" in Fig. 2 to make it a bit more clear that these are the same thing

---

> ### Author Rebuttal · Authors · 2025-07-31
>
> # [W1, Q3] Core Motivation
> For a detailed explanation of our core motivation, we kindly refer the reviewer to our response to **Reviewer Ys6A’s [W1, Q1]**, due to space constraints.
> ***
>
> # [W2] Fairness of Attention Comparison
> We understand the reviewer’s concern and recognize that our intent with Figure 1(b) may not have been fully conveyed.
> Our goal was not only to emphasize [EOS] as dominant in weight, but also to expose a deeper issue:
> **In current VTG architectures, word tokens tend to lose their grounding and act as redundant surrogates of [EOS]**.
> In Figure 1(b), we highlight the phrase **“man in gray suits”**. Although the object appears clearly in both background and foreground clips (clips 38–44), the corresponding word tokens remain largely inactive in background clips. This indicates that word tokens fail to maintain their semantic grounding and function as weak surrogates of the [EOS] token. This pattern is consistent with our controlled experiments in Figure 1(a), where excluding word tokens (“EOS only”) yields performance that matches or exceeds full-sequence input in IV2 feature.
> ***
> # [W3] Generalization of the EOS Over-Reliance
> We provide a quantitative analysis across the entire dataset to validate the prevalence of the [EOS] over-reliance phenomenon. Specifically, we measure the statistical correlation between the attention weights assigned to the [EOS] token and those assigned to the individual word tokens during cross-modal interaction. To ensure robustness, we adopt both **Pearson and Spearman correlation coefficients**—widely used metrics for evaluating linear and rank-based relationships, respectively.
>
> The Pearson correlation coefficient is given by:
>
> $$
> r_{xy} = \frac{\sum_{i=1}^{n} (X_i - \bar{X})(Y_i - \bar{Y})}{\sqrt{\sum_{i=1}^{n} (X_i - \bar{X})^2 \sum_{i=1}^{n} (Y_i - \bar{Y})^2}}
> $$
>
> Where $r_{xy}$ measures the **linear relationship** between $X$ and $Y$, with $X_i$ and $Y_i$ as the individual data points, and $\bar{X}$ and $\bar{Y}$ as the mean values of $X$ and $Y$.
>
> The Spearman rank correlation is computed as:
>
> $$
> \rho_{xy} = 1 - \frac{6 \sum_{i=1}^{n} d_i^2}{n(n^2 - 1)}
> $$
>
> Where $\rho_{xy}$ evaluates the **monotonic relationship** between $X$ and $Y$, with $d_i$ representing the **rank differences** and $n$ as the number of data points.
>
> The full measurement process is as follows:
>
> 1. **Cross-modal attention map**: For each sample in the training and validation set, we extract the cross-modal attention matrix of shape $[L,T]$, where $L$ is the number of text tokens and $T$ is the number of video clips.
> 2. **Isolate attention vectors**: We treat the attention matrix $A \in \mathbb{R}^{L \times T}$ as a set of $T$-dimensional weight arrays, and separate the [EOS] token’s attention vector $A_{EOS} \in \mathbb{R}^{T}$ and those of the remaining $L-1$ word tokens $A_{word} \in \mathbb{R}^{(L-1) \times T}$ for further correlation analysis.
> 3. **Compute token-wise correlations**: For each of the $(L-1)$ word tokens' weight arrays, we compute the Pearson and Spearman correlation with $A_{EOS}$, resulting in $(L-1)$ correlation values.
> 4. **Average correlation**: The average of the $(L-1)$ correlation scores is computed for each sample, and then the mean correlation is calculated across all samples in the subset, reporting values for both training and validation sets.
>
> | Model | Train | | Val | |  |  |  |  |
> |:---:|:---:|:---:|:---:|:---:|:---:|:---:|:---:|:---:|
> |  | **Pearson** | **Spearman** | **Pearson** | **Spearman** |  |  |  |  |
> | **CG-DETR** | 0.8960 | 0.8914 | 0.5962 | 0.7622 |  |  |  |  |
> | **TR-DETR** | 0.8110 | 0.7753 | 0.6021 | 0.6340 |  |  |  |  |
> | **FlashVTG** | 0.9745 | 0.9801 | 0.6771 | 0.7800 |  |  |  |  |
>
> A high correlation (close to 1.0) indicates that word tokens exhibit nearly identical attention patterns as [EOS].
> Prior VTG models exhibit consistently high correlation values across both Pearson and Spearman metrics, indicating that the attention distributions of word tokens closely resemble that of the [EOS] token. This suggests that word-level semantics are highly overridden, resulting in identical alignment centered around the [EOS] token in general.
> ***
> # [W4] Phrase-level Attention in DualGround
> | Model | Train |  | Val |  |
> |:---:|:---:|:---:|:---:|:---:|
> |  | **Pearson** | **Spearman** | **Pearson** | **Spearman** |
> | **DualGround** | 0.2218 | 0.2371 | 0.1185 | 0.1760 |
>
> While the updated NeurIPS policy prohibits additional visualizations in rebuttal, we instead provide a quantitative alternative to demonstrate that DualGround indeed leverages phrase-level information in a distinct manner.
>
> To address this, we compute the Pearson and Spearman correlation between the sentence-level representation $V_s \in \mathbb{R}^{T \times d}$ and the phrase-clip context vector $C \in \mathbb{R}^{N \times T \times d}$. Specifically, we first compute the L2 norm of each vector along the $d$-dimension to reduce both $V_s$ and $C$ to scalar activation maps over clips. For $C$, we obtain $N$ isolated activation maps per phrase and compute their individual correlations with the $V_s$. The average of these $N$ correlation scores is reported as the sample-level correlation.
>
> These scores are substantially lower than the [EOS]-word token attention correlations observed in baseline models (**detailed in [W3]**). While not directly comparable due to architectural differences—our model structurally separates [EOS] from word tokens—the result indicates that the two paths in DualGround exhibit compensatory behavior. This suggests that the phrase-guided mechanism helps mitigate EOS over-reliance by preserving localized semantics.
> We hope these results help clarifying the functional role of each path and support the reviewer’s understanding of how DualGround leverages phrase-level information.
> ***
> # [W5] Marginal Performance in Highlight Detection
>
> We acknoledge that the improvement in Highlight Detection (HD) is relatively marginal compared to the significant gains observed in Moment Retrieval (MR). We believe this difference stems from the fundamental nature of each task.
>
> MR involves retrieving discrete temporal segments based on semantic matching. Unlike MR, HD requires regression of highlight scores for each clip. While our dual-branch architecture enables more accurate and semantically grounded moment retrieval—evidenced by the notable improvement in R1@0.7 metrics—it poses different challenges in HD. We hypothesize that the two paths in our model capture complementary yet distinct aspects of video-text alignment, and an uncalibrated fusion of the two could introduce inconsistencies in score prediction. As future work, we plan to explore adaptive mechanisms, such as gating, to selectively balance global (sentence-level) and local (phrase-level) information for more precise and robust highlight scoring.
> ***
> # [W6] Clarifying the Dummy Encoder
>
> We thank the reviewer for pointing this out. We agree that explicitly linking f_enc in the main text with the "dummy encoder" depicted in Figure 2 would improve clarity. We will revise the manuscript to clarify this correspondence in both the figure caption and relevant textual descriptions.
> ***
> # [Q1] Discrepancy between IV2 and SF+C
>
> The performance gap between IV2 and SF+C in the [EOS]-only setting stems from differences in representation capacity and pretraining objectives:
>
> 1. **Feature dimensionality**: Text encoder of IV2 produces 4096-dimensional embeddings, offering richer sentence-level semantics than the 512-dimensional CLIP features used in SF+C.
> 2. **Pretraining domain**: IV2 is trained on video-text pairs, making its [EOS] token more aligned with temporal and action-related cues than CLIP, which is trained on image-text data.
>
> As a result, IV2 feature can rely more on [EOS], while SF+C feature benefit from token-level cues due to weaker global representations. However, we view that the stronger reliance on word tokens in SF+C models does not reflect an intentional architectural design to utilize token-level semantics. Instead, it compensates for the weaker sentence-level representation of the [EOS] token in this setting. This is further supported by the fact that our model outperforms all baselines in the SF+C setup. By explicitly modeling phrase-level semantics, our method addresses the structural limitation of existing models that rely implicitly on token-level information without architectural support.
>
> Looking forward, as pretrained video-language models continue to improve and sentence-level representations become increasingly powerful (exemplified by IV2), the tendency of VTG models to over-rely on [EOS] is likely to intensify. We believe our approach offers a principled step toward mitigating this risk in future VTG research.
> ***
> # [Q2] Phrase vs. Word-level Path
> To leverage the localized semantics of word tokens in VTG, we cluster them into coherent phrase units. This phrase-level abstraction offers two advantages over a word-level path.
>
> First, representing a sentence using N phrases (where N < L-1, the number of word tokens) reduces the computational overhead in subsequent processing while preserving semantic richness.
>
> Second, visually grounded concepts are often better represented by groupings of multiple word tokens rather than isolated words. Our phrase slots tend to cluster semantically related tokens in the feature space by residual grouping and slot attention. This often leads to tighter alignment with specific visual targets. For instance, in Fig. 1(b), tokens such as “man”, “gray”, and “suits” may individually activate across many irrelevant clips. When aggregated, their combined representation can focus attention more precisely on the intended entity, improving grounding precision.
>
> We agree that a direct comparison would further validate the effectiveness of our phrase-based design. We appreciate the reviewer’s suggestion and will explore this direction in future work.

---

> ### Comment · Reviewer_sNK1 · 2025-08-05
>
> Thank you to the authors for the detailed response. The clarification of several points is helpful. The analysis of the correlations is useful. I agree with the authors that it is not possible to directly compare the correlation values for the representation vectors with the correlations of the attention values in previous methods, but the analysis is appreciated nonetheless, and it does help to show the surrogate behaviour of word tokens in previous methods.
>
> I appreciate the **response to Q1** on the feature comparison. On this, while I see where the authors are coming from by saying that as sentence level representations improve, the models will become more reliant on [EOS] tokens, I think it's perhaps a bit pre-emptive to say that models will become overreliant. It's also possible that with improved representations, the [EOS] token will become sufficient and word-level representations will not be necessary anymore. But this all remains to be seen.
>
> I have come to agree somewhat with reviewer Ys6A that it is debatable that the use of the [EOS] token is inherently a flaw. If a model is able to use the [EOS] token to get good results, then evidently there is enough information in the [EOS] token. The word tokens may become surrogates of the [EOS] token because they are redundant. However, if one desires to understand the individual components of the sentence, then phrase/word level knowledge would be desired. As it stands, the model is able to improve upon existing methods by utilising an approach which explicitly invokes phrase-level knowledge. As such, this phrase-level knowledge is useful here. I think the usefulness is highlighted particularly by the impact of sentence length  as in Table 4 - in longer sentences where it is more difficult to represent the full sentence with the [EOS], the phrase-level knowledge becomes more important.
>
> **For the response to Q2**, I appreciate that at a phrase-level it is possible to indicate more specific concepts than it is at a word-level. However, there is perhaps something to say here in that you could arguably say the same about phrase-level vs sentence-level - at a sentence-level you have more context and information in the representation that allows the method to focus more on the specific correct moment. It is highlighted in Figure 1(b) how the "man in gray suits" tokens are not attended to despite there being a man in a gray suit in the video, and this is deemed a fault of the previous models, implying that attention should be high in those other moments that are not relevant to the overall query itself. However, now at a word-level, it is said that high attention levels for "man" or "gray" at other points outside the moment are a problem, as this distracts from the specific desired moment. This implies that attending correctly to the individual components is a problem, rather than a benefit.
>
> Likewise, it could be said that if there were other men in gray suits in the video, then it would be problematic for the phrase-level token to attend to the parts of the video containing those other men. This ultimately leads to the end point that it is better to utilise the sentence-level token, as this will contain all the relevant context for the moment and won't attend to the irrelevant parts of the video. Maybe there is a sweet spot for this and maybe it is at a phrase-level, perhaps related to the prevalence of word-level entities vs phrase-level entities in a given video, but I thought I should point out this logic, as it seems that by following it you can potentially contradict the motivation of the paper.
>
> While I still have some concerns over the motivation, I do still believe the method is useful and the results are sufficiently strong. I currently intend to retain my rating.

---

> > ### Author Response · Authors · 2025-08-05
> >
> > Thank you for the constructive and thoughtful review, and for recognizing the usefulness of our method. We appreciate your positive remarks on our correlation analysis, the clarification of experimental details, and the defense of our design choices in [Q2] and [W4]. Your feedback provides an opportunity to further clarify our motivation and address the points you raised.
> >
> > **"Our motivation is not to argue that relying on [EOS] is inherently flawed, but rather that it is not sufficient in all cases. Therefore, we introduce a complementary phrase-level path to address the limitations of unguided global representations."**
> > The current [EOS], derived solely from a pretrained text encoder without visual grounding, can underrepresent visually salient cues or complex sentence semantics, which may lead to missed grounding opportunities.
> >
> > We provide empirical evidence in Supplementary Fig. A2, second sample (“A guy with red jacket is getting interviewed”). Prior VTG models, which rely heavily on the [EOS] token, and the sentence path of DualGround (refer to (d)) retrieved an interviewing scene while missing the “red jacket” cue, whereas the phrase path (refer to (c)) successfully localized the correct moment by preserving and utilizing local semantics.
> >
> > A consistent pattern is observed in the third sample “Woman flexes her muscles for photos”, where the [EOS] token, with higher emphasis on “photos,” caused prior models and our sentence path to also focus on an early photo-taking scene. In contrast, the phrase path concentrated on “muscles,” successfully grounding the correct moment. These cases illustrate how preserving localized semantics can complement the global representation of [EOS] to improve grounding in challenging scenarios.
> >
> > In addition to clarifying our motivation, we would also like to address the concern regarding a potential inconsistency between Figure 1(b) and our [Q2] response. In Figure 1(b), our intent was to show that word tokens in prior VTG models often act as [EOS] surrogates with minimal independent activation, which may indicate a tendency toward over-reliance. In contrast, our Q2 discussion focused on cases where individual word activations can become noisy and attend to irrelevant instances, which can be mitigated by aggregating them into phrase-level units for more precise focus.
> >
> > We agree that, just as a phrase-level representation can offer richer semantics than individual words, a sentence-level representation can often provide more complete context than a phrase-level representation. However, the [EOS] token does not always serve as an ideal sentence-level representation, as it can underrepresent localized elements that are visually important for grounding. Our dual-path design addresses this by letting the phrase path preserve such local semantics, while still leveraging the global context from the sentence path to ensure robustness across scenarios.

---

> > > ### Comment · Reviewer_sNK1 · 2025-08-07
> > >
> > > Thank you for the response. I agree in principle that incorporating local semantics is useful where the global semantics do not sufficiently represent the meaning. I think there may just be a trade-off or a grey area with regards to the local representations highlighting irrelevant sections of the video. While examples are presented where the phrase path has contributed positively when combined with the sentence path, I imagine there may be other examples where the sentence path is successful but the phrase path contributes negatively by highlighting an only partially relevant video segment (in the example this could be a different man in a gray suit, or the same man in the gray suit but at a different point in the video). I can't verify this myself of course, but in the red jacket example in Figure A2 for example, the phrase-level path shows higher activations at earlier points in the video that don't seem relevant.
> > >
> > > My point was mainly in response to the claim that word-level representations would highlight irrelevant moments relative to phrase-level representations, and I wanted to point out that the same could be said between phrase-level and sentence-level. So this is maybe a word of caution against using that reasoning for using phrase-level as opposed to word-level. It's not that I disagree on using phrase-level, I just think the reasoning may need to be thought through. I think it is likely that there is a sweet spot for the granularity of the local representations that incorporates local semantics without distracting too much with irrelevant moments, but it is not clear what that sweet spot is. It may also depend on how well the sentence-level path understands the semantics. Maybe the better the [EOS] representation, the more fine-grained the local representation should be. Maybe this is something that could be worth thinking about for future works.
> > >
> > > I think regardless of the finer details around the motivations of the phrase-level path, the method seems thought through and the results are strong enough. So overall, the responses have addressed my concerns.

---

> > > > ### Author Response · Authors · 2025-08-08
> > > >
> > > > We sincerely appreciate the reviewer’s thoughtful follow-up and the balanced perspective provided in this discussion. The careful articulation of potential trade-offs between phrase-level and sentence-level representations demonstrates a deep understanding of the core design considerations in our work.
> > > >
> > > > We agree with the reviewer that while phrase-level representations can capture localized semantics that sentence-level representations may overlook, the opposite can also occur—phrase-level units may occasionally highlight partially irrelevant moments. As also noted in our [W5] response, the two paths in our model capture complementary yet not fully consistent aspects of video-text alignment. This can be beneficial for diverse scenarios in moment retrieval but may, in some cases, lead to minor inconsistencies in tasks such as highlight detection. To mitigate (though admittedly not entirely eliminate) this risk, our model applies an EOS reconstruction loss that encourages aggregated phrases to maintain semantic consistency with the sentence representation, and guides the phrase path to focus on clips that are also meaningful from the sentence-level perspective.
> > > >
> > > > We believe the value of our work lies in diagnosing a concrete limitation in the cross-modal interaction process of prior VTG models—a process that is central to video-text tasks—and proposing a new dual-path architecture to address it. As the reviewer notes, effectively informing local semantics while jointly regulating their interaction with global context is a promising and important direction for future research, and we fully agree on its merit.
> > > >
> > > > We again thank the reviewer for the constructive insights and for recognizing the strengths of our proposed approach.

---

> > > > > ### Comment · Reviewer_sNK1 · 2025-08-08
> > > > >
> > > > > Thank you for the discussion and for addressing my concerns. As a final point, perhaps another way to contrast the abilities of the models reliant on sentence-level representation with the joint phrase-level and sentence-level method could be to present the models with some slightly altered queries (e.g. in the red jacket example you could use "green jacket" instead of "red jacket") and show the compared model outputs. The previous methods may not show a significant change while ideally the DualGround approach would show a change due to the altered fine-grained semantic meaning.

---

### Official Review · Reviewer_Ys6A · 2025-07-02

**Clarity:** 2
**Significance:** 2
**Originality:** 2
**Rating:** 4
**Confidence:** 3

**Summary:**

This paper argues that Video Temporal Grounding (VTG) models over-rely on the global [EOS] token, which neglects word-level cues and hinders fine-grained localization. To address this, it proposes DualGround, a dual-branch architecture that disentangles semantics. A Sentence-Level Path handles global alignment using the [EOS] token, while a Phrase-Level Path clusters words to capture local context. Fusing these paths aims for more accurate grounding, validated by state-of-the-art results on multiple benchmarks.

**Questions:**

### 1. On the Fundamental Motivation
The paper's motivation hinges on the premise that relying on the [EOS] token is a flaw. However, since the [EOS] token is trained to encapsulate the entire sentence's meaning, one could argue it is the ideal representation for a sentence-level task like VTG.
- Could you please elaborate on why and in which specific scenarios "fine-grained temporal alignment" is critical? Providing a concrete example with sequential actions (e.g., "A man picks up a ball and then throws it"), where the [EOS] token alone would be insufficient, would greatly strengthen your motivation. A convincing answer here is critical. If the authors can prove that a significant portion of the VTG task requires more than global semantics, my assessment of the work's Significance would increase substantially.

### 2. On the Phrase-Level Path's Mechanism
The effectiveness and interpretation of this core contribution are unclear, which impacts the perceived novelty and quality of the work.
- The paper claims to group words into "semantically coherent phrases," but visualizations suggest the model attends to individual keywords. Could you clarify this discrepancy? Is the goal to form syntactic phrases or to disentangle core concepts?
- Your analysis shows the optimal $N$ depends on query length, suggesting the fixed-$N$ approach is a limitation. Have you considered methods for dynamically adapting $N$ based on query complexity? A discussion on their potential advantages or disadvantages would be insightful. Clarifying these points is essential for understanding your main technical contribution. A satisfactory response would significantly improve my assessment of the paper's Originality and Quality.

### 3. On the Sentence-Level Path's Mechanism
The paper claims that dummy tokens act as "semantic attractors" for irrelevant content, which is an interesting architectural choice.
- Could you provide a visualization (e.g., attention maps) to support this claim? Showing how irrelevant video clips attend more to dummy tokens while relevant clips focus on the [EOS] token would be necessary to validate this mechanism. Providing this evidence would increase my confidence in the paper's architectural Quality.

### 4. On Experimental Clarity and Justification Several details are needed to fully assess the validity of the experiments.
- For the "Ours" model in Figure 1(a), what was the exact configuration for the "EOS only" condition? Also, given that compared models have different architectures, is it fair to conclude the universal importance of word-level semantics from this experiment?
- What is the exact architecture for the baseline setting (a) in Table 3? Please confirm if it is a model composed solely of the sentence-level path. These clarifications are necessary for reproducibility. Answering them satisfactorily would increase my confidence in the experimental results and thus the paper's overall Quality.

### 5. Positioning within Related Work on Holistic Understanding
"Let Me Finish My Sentence: Video Temporal Grounding with Holistic Text Understanding" (Woo et al., MM '24), addresses the importance of holistic query understanding in VTG. While your paper separates global ([EOS]) and local (phrase) semantics, Woo et al. use a global text anchor to create a gating mechanism.
- Could you discuss how your DualGround approach is positioned relative to this work? Specifically, what are the conceptual advantages and disadvantages of your dual-path separation compared to their global anchor-based gating mechanism? Situating your work in relation to highly relevant, concurrent research is crucial for clarifying its unique contribution. A thoughtful comparison would significantly strengthen the "Related Work" section and my assessment of the paper's Originality.

**Ethical Concerns:**

["NO or VERY MINOR ethics concerns only"]

**Final Justification:**

The rebuttal clarifies many of the concerns in the original review. I raise my rating from 3 to 4.

**Limitations:**

Yes

**Quality:**

2

**Strengths And Weaknesses:**

### Strengths
- The proposed DualGround architecture is an intuitive solution. Its separation of global and local semantics is a well-motivated design that directly addresses the identified problem.
- The method is validated with extensive experiments, achieving consistent state-of-the-art results across multiple benchmarks.

### Weaknesses
- Questionable Core Motivation: The paper's central premise—that reliance on the [EOS] token is a flaw—is debatable. Since VTG is a sentence-level task, using a token designed to represent the entire sentence seems logical, not problematic. The paper needs a stronger justification for why this is a fundamental limitation.
- The central claim of learning "semantically coherent phrases" is weakly supported. In visualizations like Figure 3(e) and Figure A2, the model appears to focus on individual keywords (e.g., 'man', 'speaking', 'train') rather than composing meaningful phrases (e.g., "A man in black"). This raises doubts about whether the phrase-level path is truly learning phrasal semantics or is acting more like a keyword-weighting mechanism.
- The use of a fixed $N$ for all queries, regardless of their length or complexity, is a major limitation. The authors' own analysis in the supplementary material (Sec. C), which shows that the optimal $N$ varies by dataset based on average query length, ironically argues against a fixed $N$ and highlights the need for a dynamic approach.
- While the overall architecture of DualGround is original, several of its core components are adapted from prior work. For instance, the ACA in the sentence-level path is from CG-DETR, the decoding path is from FlashVTG, and the idea for phrase generation is inspired by LGI. This slightly diminishes the paper's originality, as its main contribution lies more in the creative combination and design of effective existing modules rather than the invention of entirely new ones.

---

> ### Author Rebuttal · Authors · 2025-07-30
>
> # [W1, Q1] EOS Reliance and Motivation
> A core motivation behind our work is not simply that attention concentrates on the [EOS] token, but that word tokens become functionally indistinguishable from [EOS] during cross-modal alignment. This phenomenon arises from two factors in prior VTG models.
>
> **1. Shared Projection for Cross-Modal Interaction:**  Most VTG models use a shared projection layer for all text tokens before cross-modal attention. As [EOS] is explicitly trained to represent global semantics, it becomes the dominant alignment target during training.
>
> **2. Semantic Collapse:**  Since grounding loss can be optimized by aligning [EOS] with relevant video clips, word tokens are implicitly drawn toward [EOS] in embedding space. Their attention patterns converge, leading to semantic redundancy. (**For further clarification, please refer to Reviewer sNK1-[W3]**)
>
> This convergence results in a semantic collapse where word tokens lose their individual meaning, acting as redundant surrogates of [EOS]. However, [EOS] is derived solely from the pretrained text encoder without visual grounding and may **overlook key visual cues** such as objects or actions. This limitation undermines accurate grounding in cases where localized semantics are essential.
>
> Our model overcomes this challenge by leveraging both the strong sentence-level representation of the [EOS] token and the local semantics of the individual words. While [EOS] is still important for global understanding, our approach ensures that word-level semantics are preserved to localize precise moments that may be missed when relying solely on [EOS] for alignment.
>
> The above problem and our contribution are evident in Supplementary Fig. A2. In the query “A guy with red jacket is getting interviewed,” both baselines and DualGround’s sentence path (d) retrieve the interview scene but miss the visual cue “red jacket”. The phrase path (c), however, focuses on the correct clip and retrieves the precise moment. Similarly, third sample in Fig. A2 “Woman flexes her muscles for photos,” baseline models and sentence-level predictions attend to the photo-taking scene, while the phrase path correctly identifies the moment of muscle flexing.
> To support this claim, we examine the attention weights of [EOS] token from InternVideo2 final self-attention layer. The tokens **“red jacket”** and **“muscles”** receive low attention scores, indicating that [EOS] underrepresents visually grounded content, especially when such cues are critical for localization.
>
> | index | 0 | 1 | 2 | 3 | 4 | 5 | 6 | 7 | 8 | 9 | 10 | 11 | 12 |
> |:---:|---|---|---|---|---|---|---|---|---|---|---|---|---|
> |**token**| [sos] | A | guy | _y | with | **red** | **jack** | **_et** | is | getting | **interview** | **_ed** | [eos] |
> | **[eos]** | 0.0908 | 0.0007 | 0.1205 | 0.0366 | 0.0083 | **0.0068** | **0.0037** | **0.0148** | 0.0284 | 0.0232 | **0.1064** | **0.2633** | 0.2959 |
>
> | index | 0 | 1 | 2 | 3 | 4 | 5 | 6 | 7 | 8 | 9 |
> |:---:|---|---|---|---|---|---|---|---|---|---|
> | **token**| [sos] | Woman | flex | _es | her | **mus** | **_cles** | for | **photos** | [eos] |
> | **[eos]** | 0.0632 | 0.0429 | 0.0165 | 0.0906 | 0.0161 | **0.0961** | **0.0669** | 0.0301 | **0.2668** | 0.3103 |\
> ***
>
> # [W2, Q2] Interpretation of Phrase
>
> Our use of the term **“semantically coherent phrases”** may have suggested the formation of linguistically well-formed expressions (e.g., "a man in black"). However, we would like to clarify intended meaning.
> Our approach focuses on grouping semantically related (**i.e semantically coherent**) word tokens based on their representational similarity in the feature space. Through our RPG and Slot Attention modules, each slot attends to a subset of tokens that tend to represent related concepts. While it is true that the visualizations suggest the model appears to focus strongly on individual keywords (e.g., 'man', 'speaking', 'train'), we interpret this behavior as a natural outcome of the model's tendency to first anchor on a core token, with surrounding words providing complementary context.
>
> We recognize that in some cases, the phrase-level path may resemble keyword weighting, where individual word tokens dominate attention. However, this behavior is not unintended; it reflects the way the model clusters semantically related tokens in the feature space, leading to a **soft clustering** of related concepts rather than rigid syntactic units. This explains why certain words receive stronger attention, with the meaning of nearby surrounding words being integrated naturally to form coherent phrase-level representations.
>
> In this sense, while the model’s behavior may sometimes appear similar to keyword weighting, the goal of grouping semantically related words into phrases is still upheld, and the representation effectively captures the relevant semantics for moment retrieval tasks.
>
> ***
> # [W3, Q2] Static N Constraint
> We agree that highlighting the limitations in our work is important. This question overlaps with **Reviewer 3BT3-[W1]**, and due to space constraints, we kindly refer the reviewer to our response to **Reviewr 3BT3-[W1]**.
>
> ***
> # [W4] Module Originality Concern
>
> We acknowledge that core components in our design are inspired by prior works (e.g., LGI, CG-DETR, FlashVTG).
> However, our aim was not to simply reuse existing modules, but to structurally reframe them within a novel dual-path architecture that mitigates their individual limitations. During phrase grouping, we incorporate position embeddings to reflect the sequential and contextual nature of linguistic phrases, an aspect often overlooked in prior approaches. We then apply Slot Attention to enhance semantic coherence and guide the resulting phrase representations toward the sentence intent via a reconstruction loss. These design choices collectively contribute to a more interpretable representation, which is central to our dual-path architecture.
> ***
> # [Q3] Dummy Token as Attractor
>
> While we cannot include additional visualizations, we provide a theoretical explanation of the role of dummy tokens as **“semantic attractors”**.
>
> In our sentence-level path, we introduce an auxiliary loss that regresses the attention weight assigned to the [EOS] token toward the highlight detection (HD) score of each video clip. (**This is outlined in Section 3.5 HD Loss**) Given that HD scores reflect the semantic relevance of each clip with respect to the input query, this supervision encourages the model to assign higher attention to the [EOS] token for salient clips. In contrast, non-salient clips—those with low HD scores—are not encouraged to attend to [EOS] and thus distribute their attention toward the dummy tokens.
> As a result, dummy tokens function as attractors for irrelevant content.
> ***
> # [Q4] On Experimental Clarity and Justification
> 1. **"EOS only" in Figure 1(a)**:
>    This variant uses only the sentence-level path with the [EOS] token as input. The phrase-level path is completely disabled.
>
> 2. **Purpose of the experiment**:
>    The goal of this ablation is to justify the observed underutilization of word tokens in standard VTG architectures. More direct discussion of word semantics and their contribution to performance is provided in our response to **[W1, Q1]**.
>
> 3. **Baseline (a) in Table 3**:
>    This setting uses the sentence-level path with the full token sequence (i.e., including all word tokens and the [EOS] token). We acknowledge that this was not clearly distinguished from the "EOS only" setting in Figure 1(a), and we will revise the manuscript for clarity.
> ***
> # [Q5] Comparison to Holistic Query Models
>
> **Conceptual Overlap:**
> Woo et al.[1] introduces a global anchor via mean pooling over text tokens, which gates noisy frames. This aligns conceptually with our Adaptive Cross Attention (ACA), where [EOS]—serving as a sentence-level anchor—is used to inform dummy tokens that absorb attention from less relevant clips.
>
> **Key Architectural Differences:**
> Unlike the approach of Woo et al., our design explicitly separates global and local semantics. We use the [EOS] token as the sole sentence-level representation and apply grouping modules to remaining word tokens into phrase-level clusters. This results in a dual-path architecture that treats sentence and phrase-level semantics as structurally distinct, enabling finer control and interpretability.
>
> This distinction leads to several advantages.
> - **Interpretable Design with Localized Cue Awareness**:
>    By separating [EOS] and word tokens into global and local paths, our model preserves token-level distinctions and captures localized visual cues, leading to stronger interpretability and fine-grained video-text interactions.
>
> - **Direct Gating Supervision**:
>    Unlike Woo et al., our ACA’s attention weights are explicitly supervised using highlight scores, encouraging alignment between sentence relevance and video importance in a more grounded and controllable manner.
>
> - **Stability in Cross-Modal Matching**:
>    Using a dummy token for gating provides a discrete mechanism to filter irrelevant clips, whereas feature-wise gating based on continuous similarity scores can be more sensitive to alignment noise between video and text embeddings.
>
> However, we acknowledge trade-offs in this approach.
> - **Computation Overhead**:
>    The dual-path architecture introduces additional components, increasing model complexity and training cost.
>
> - **Semantic Granularity Gap**:
>    While the phrase path enhances fine-grained localization by focusing on localized semantics, this path may diverge from the globally consistent representation that highlight detection relies on. As a result, its influence on clip-wise saliency scoring can be limited.
>
> [1] Woo, Jongbhin, et al. "Let me finish my sentence: Video temporal grounding with holistic text understanding." Proceedings of the 32nd ACM International Conference on Multimedia. 2024.

---

> > ### Comment · Reviewer_Ys6A · 2025-08-05
> >
> > The "semantic collapse" is predicated on a questionable assumption that the [EOS] token fails to encode critical visual cues (e.g., "red jacket") in modern Vision-Language Models, and the supporting evidence (final-layer attention) is insufficient to prove it. Similarly, the key architectural claim in [Q3] is unverified, as the provided theoretical explanation cannot replace the requested empirical proof.
> >
> > On the other hand, the rebuttals for [Q2] and [Q4] were useful, clarifying experimental ambiguities and defended key design choices. However, because the paper's foundational claims in [Q1] and [Q3] remain unsubstantiated, the initial rating is maintained.

---

> ### Author Response · Authors · 2025-08-05
>
> Thank you for the detailed review and for pointing out your concerns regarding [Q1] and [Q3]. I would like to clarify our intention and provide additional context for both points.
>
> ## [Q1]
> **Our argument is not that the [EOS] token in modern VLMs completely fails to encode such cues, but that in certain scenarios its aggregated representation can underrepresent them, leading to missed groundings.**
> To support this, we presented not only the final-layer attention weights from the pretrained text encoder but also the moment retrieval results of prior VTG models comparing with ours, their highlight detection scores, and the activation patterns of our dual-branch model for two specific samples (second and third samples in Supplementary Fig. A2).
> We would like to note that the provided evidence includes multiple sources that collectively illustrate the **"EOS token does not always provide an ideal sentence-level representation for VTG"**.
>
>
> ## [Q3]
> NeurIPS rebuttal policy this year prevented us from adding new visualizations, so we substituted them with a detailed theoretical explanation describing how dummy tokens act as “semantic attractors”.
> In our sentence-level path, the HD loss supervises the [EOS] attention weights to match the highlight detection scores, encouraging salient clips to attend more to [EOS] while non-salient clips distribute attention toward dummy tokens. This mechanism directly follows from the design of the supervision and its intended effect. As such, we are not certain why this was considered “unverified,” given that the described behavior is a direct consequence of the loss formulation in Section 3.5.
>
> We hope this clarification helps in understanding our reasoning for Q1 and Q3, and we appreciate your consideration.
> If you have any remaining concerns or questions, we would be happy to clarify and further discuss. We're open to communication and believe there is still sufficient time for constructive dialogue.

---

> > ### Comment · Reviewer_Ys6A · 2025-08-08
> >
> > Since some of the concerns are resolved, I will raise my score from 3 to 4. Please include the clarifications in the final version if the paper is accepted.

---

> > > ### Author Response · Authors · 2025-08-09
> > >
> > > Thank you for your thoughtful follow-up and for reconsidering your rating.
> > > We truly appreciate the depth of your analysis and the professionalism you brought to your review.
> > >
> > > We will make sure to incorporate the clarifications and improvements discussed into the final version.

---

### Comment · Area_Chair_QrxB · 2025-08-05

Dear Reviewers,

Thank you very much again for performing this extremely valuable service to the NeurIPS authors and organizers.

As the authors have provided detailed responses, it would be greatly appreciated if you could take a moment to review them and see if your concerns have been addressed. Given that the discussion phase is nearing its end, your prompt feedback would be especially valuable, allowing the authors a final opportunity to offer any additional clarifications if needed.

Cheers,

AC

---

### Note · Authors · 2025-08-12

We sincerely thank all reviewers for their insightful and constructive feedback throughout the review and discussion phases. The thoughtful comments and questions have greatly helped us refine our explanations, clarify the motivations behind our design choices, and better position our contributions within the broader VTG research landscape.

Our work quantitatively and qualitatively examines the phenomenon of [EOS] over-reliance in prior VTG models, providing analyses of cases where this reliance can negatively impact temporal grounding. We present both statistical evidence and qualitative examples to illustrate the issue, and we propose a dual-path architecture that jointly leverages sentence-level and phrase-level representations to mitigate these limitations. This approach aims to capture complementary aspects of global and local semantics, offering a more robust framework for cross-modal interaction.

We deeply appreciate the reviewers’ discussions on the potential trade-offs between sentence-level and phrase-level representations, as well as their suggestions for exploring adaptive mechanisms and alternative evaluation settings. We acknowledge that balancing the complementary strengths of global and local semantics is a challenging and valuable direction, and we will incorporate the clarifications, additional analyses, and suggested experimental ideas into the final version and into our future research agenda.

Finally, we would like to express our gratitude to the reviewers and the AC for engaging with our work in such a constructive and responsible manner. The feedback provided during the rebuttal and discussion phases has been invaluable in strengthening both the clarity and the rigor of our paper, and we believe it will positively influence our future research in this area.

---

### Decision · Program_Chairs · 2025-09-17

**Decision:**

Accept (poster)

**Comment:**

This authors propose to disentangle textual semantics to address the problem of over-reliance on EOS token, which results in limited fine-grained localization. This work demonstrates intuitive solutions and validate them with extensive experiments and ablation studies. Even though there are several concerns regarding illustrative figures and technical complexities, the authors elegantly handle them. In overall, I recommend this paper being accepted.